# Resolving colistin resistance and heteroresistance in *Enterobacter* species

Swapnil Prakash Doijad[1,2,22], Nicolas Gisch [3,22], Renate Frantz[1,2], Bajarang Vasant Kumbhar[4], Jane Falgenhauer[1,2], Can Imirzalioglu[1,2], Linda Falgenhauer[1,2,5], Alexander Mischnik [1,6], Jan Rupp [1,6], Michael Behnke[1,7], Michael Buhl[1,8,9,10], Simone Eisenbeis[1,9], Petra Gastmeier[1,7], Hanna Gölz[1,11], Georg Alexander Häcker[1,11], Nadja Käding[1,6], Winfried V. Kern [1,12], Axel Kola [1,7], Evelyn Kramme[6,7], Silke Peter[1,8], Anna M. Rohde[1,7], Harald Seifert [1,13], Evelina Tacconelli[1,9], Maria J. G. T. Vehreschild[1,14,15], Sarah V. Walker [1,13], Janine Zweigner[1,13], Dominik Schwudke [1,3,16], DZIF R-Net Study Group* & Trinad Chakraborty [1,2]✉

Species within the *Enterobacter cloacae* complex (ECC) include globally important nosocomial pathogens. A three-year study of ECC in Germany identified *Enterobacter xiangfangensis* as the most common species (65.5%) detected, a result replicated by examining a global pool of 3246 isolates. Antibiotic resistance profiling revealed widespread resistance and heteroresistance to the antibiotic colistin and detected the mobile colistin resistance (*mcr*)−9 gene in 19.2% of all isolates. We show that resistance and heteroresistance properties depend on the chromosomal *arnBCADTEF* gene cassette whose products catalyze transfer of L-Ara4N to lipid A. Using comparative genomics, mutational analysis, and quantitative lipid A profiling we demonstrate that intrinsic lipid A modification levels are genospecies-dependent and governed by allelic variations in *phoPQ* and *mgrB*, that encode a two-component sensor-activator system and specific inhibitor peptide. By generating *phoPQ* chimeras and combining them with *mgrB* alleles, we show that interactions at the pH-sensing interface of the sensory histidine kinase *phoQ* dictate *arnBCADTEF* expression levels. To minimize therapeutic failures, we developed an assay that accurately detects colistin resistance levels for any ECC isolate.

Members of the order *Enterobacterales* are a frequent source of morbidity and mortality associated with bloodstream, respiratory tract, and urinary tract infection in healthcare institutions worldwide[1,2]. Among the *Enterobacterales*, *Enterobacter* are opportunistic nosocomial pathogens listed among the top five species causing bloodstream infections (BSI)[3–5], and are in WHO's priority list for which new antimicrobials are urgently needed[6]. *Enterobacter* are also present in

environmental habitats such as soil and water and exhibit considerable phenotypic and genomic diversity[7]. As a result of taxonomical complexity, pathogenic species are commonly grouped within the *Enterobacter cloacae* complex (ECC)[8]. Infections with *Enterobacter* species are difficult to treat as they often exhibit resistance to penicillin, quinolones, and third-generation cephalosporins[9]. While carbapenems are among the most attractive therapeutic options, *Enterobacter*

A full list of affiliations appears at the end of the paper. *A list of authors and their affiliations appears at the end of the paper.
✉e-mail: Trinad.Chakraborty@mikrobio.med.uni-giessen.de

species have emerged as the third most common hosts for carbapenemases worldwide forcing the use of colistin as a "last-resort" antibiotic for the treatment[10–12].

Colistin and other cationic peptides are membrane-active agents that target the lipopolysaccharide (LPS) component of the bacterial outer membrane (OM)[13]. Colistin is often considered a last-line therapeutic option for treating *Enterobacter* infections, but the prevalence of resistance among *Enterobacter* species is uncertain. Minimum inhibitory concentration (MIC) determinations show high variability depending on the test methods used[14]. The WHO Global Antimicrobial Resistance Surveillance System (GLASS) reports *Enterobacter* species as being naturally susceptible to colistin[15,16]. Other studies indicate that resistance towards colistin is either rare or can comprise up to 17% of all *Enterobacter* species[17,18]. A recent systematic study of *Enterobacterales* found that ~20% of *Enterobacter* isolates were colistin resistant, as compared to <2% isolates for *E. coli* and *Klebsiella*[17].

*Enterobacter* isolates frequently exhibit heteroresistance towards colistin making accurate resistance testing difficult[19,20] leading to a high risk of treatment failures, particularly with those isolates that have previously been classified as susceptible[19,21]. Previous studies have implicated the two-component systems (TCSs) PhoPQ/PmrAB that regulate expression of the *arnBCADTEF* gene cassette[22,23], the PhoPQ inhibitor- MgrB and enhancer- Ecr peptides[24], the inner membrane protein DedA[24], and AcrAB-TolC efflux pump[25] with colistin heteroresistance in different species of *Enterobacter*. Recent studies indicate that the TCS PmrAB is not involved in heteroresistance towards colistin in *E. cloacae*[20,22]. In addition, taxonomic conflicts in ECC have further complicated efforts to consistently identify genetic features underlying heteroresistance at the species level. Membership based on a *hsp60* gene classification scheme first provided clues for distinct cluster-dependent heteroresistance levels and sensitive populations within ECC[20,26]. However, heteroresistance frequencies varied greatly within specific clusters suggesting mutations in additional genes or allelic differences underly these findings.

In this study, we performed a genome-based taxonomic study on clinical isolates of *Enterobacter* obtained over a 3-year period from six university hospitals at different locations in Germany. Our data suggest temporal and geographic predominance of the species *E. xiangfangensis*, a finding that was replicated by examining whole genome sequences of 3246 isolates from clinical sources worldwide. In contrast to previous reports, we find that colistin resistance and heteroresistance in the genus *Enterobacter* are widespread, involving over 80% of all known species. As the problem of heteroresistance and its impact on unexplained treatment failures of clinical infections are of great concern, we applied phylogenomics, antimicrobial susceptibility testing, lipid A profiling together with isogenic deletion mutation and complementation analysis to reveal mechanisms that characterize colistin heteroresistance in *Enterobacter*.

## Results

### *Enterobacter xiangfangensis* lineage-1 as the most common *Enterobacter* in clinical samples

Based on MALDI-TOF and VITEK data, the majority (86.0%) of the isolates were identified as *E. cloacae*, 8.4% as unknown *Enterobacter* species, 3.0% as *E. asburiae*, and 1.2% as *E. cloacae* complex, a single isolate each as *E. cancerogenus* and *E. ludwigii*. To study taxonomic assignments in greater details, we sequenced and studied one-third of isolates from each center (n = 165, 96 from BSI and 69 from body sites) (Supplementary Data 1b).

We calculated the "overall genome relatedness index" (OGRI) of isolates by comparing them to type strains of 23 valid *Enterobacter* species as of June 2022 (Supplementary Data 5). OGRI-based genome analysis identified 12 *Enterobacter* species from 165 isolates. Of these, a total of 108 (65.5%) were identified as *E. xiangfangensis*, 23 (13.9%) as *E. ludwigii*, and 9 isolates each as either (5.4%) *E. kobei* or

*E. roggenkampii* (Supplementary Table 1). The species *E. bugandensis*, *E. cloacae*, *E. asburiae*, *E. chengduensis*, *E. mori*, *E. wuhouensis*, *E. cancerogenus*, and *E. vonholyi* were observed only sporadically (≤5 isolates each) (Fig. 1a). In total, 67 (69.8%) of 96 isolates from BSI were *E. xiangfangensis*, followed by *E. ludwigii* (12.5%) (Supplementary Table 2, Supplementary Data 1).

Core genome-based phylogenomics analysis together with Bayesian Analysis of Population Structure (BAPS) distributed *E. xiangfangensis* isolates into four major lineages, which we assign as Lineages (L)-1 to -4 (Fig. 1b). Isolates clustered consistently in four lineages irrespective of phylogenomic approaches used (see Supplementary Fig. 1). A total of 64 of 108 *E. xiangfangensis* observed (59.2%) were members of L-1 isolates, while L-2, L-3, and L-4 were represented by 26, 11, and 7 isolates, respectively.

We used the OGRI-based approach to clarify the taxonomy of the publicly available genomes based on high-quality assemblies of *Enterobacter* from 67 countries. A total of 2233 from 3246 (68.7%) isolates deposited as ECC or *Enterobacter* species were identified as *E. xiangfangensis* (Supplementary Data 2). Country-wise analysis (comprising 35 countries for which more than five genome sequences were available) indicated that *E. xiangfangensis*, particularly isolates of L-1, was the predominant species detected in 29 countries and comprised between 61 to 100% of all *Enterobacter* isolates reported therein (Supplementary Table 3).

MLST typing further sub-classified the members of each species (see Fig. 1b and Supplementary Data 1b). The most abundant "*E. cloacae*" MLST types were ST-50 and ST-190, both associated with the species *E. xiangfangensis*, L-1. Further analysis of individual MLST clones showed differences of <70 nucleotides while the pan genome differed by between 24 to 385 genes (particularly of ST-50, ST-190, ST-108, and ST-175) indicating recent spread from a common ancestor (see Supplementary Data 7). These closely related isolates of specific MLST type were obtained from different centers and at different times suggesting broad distribution and persistence of commonly occurring clones throughout Germany.

### High occurrence of *mcr-9* gene in *E. xiangfangensis*

All of the 108 *E. xiangfangensis* isolates encoded resistance genes for ß-lactams (cephalosporins) and phenicols class of antibiotics. A further 92% and 91% of *E. xiangfangensis* isolates carried genes encoding quinolone and fosfomycin resistance, respectively. Aminoglycoside, sulfonamide, and trimethoprim resistance-conferring genes were present in <25% isolates, while tetracycline, rifamycin, carbapenem, streptothricin, and bleomycin were rare (<10%). Interestingly, 9 (8.3%) *E. xiangfangensis*, 2 *E. ludwigii* and a single isolate of *E. kobei* and *E. chengduensis* each, which carried the mobile colistin resistance (*mcr*)-9 gene, were identified (Fig. 1b). Presently, an average of 2.5% (0.1–5.1%) of clinical isolates obtained from various infections are reported to carry *mcr* alleles[27] (Supplementary Table 4). This contrasts with the relatively high (8.3%) number of clinical *E. xiangfangensis* with the *mcr-9* gene in this study. Analysis of worldwide isolates revealed a total of 24.8% of *E. xiangfangensis* isolates harbor the *mcr-9* gene, while the presence of other *mcr* alleles (*mcr-1, 3, 4*, and *10*) was rare (up to 1.4%) (Supplementary Data 2 and Supplementary Fig. 2).

We explored the genetic environment by sequencing ten *mcr-9* carrying isolates to completion. For seven *E. xiangfangensis* isolates, the *mcr-9* gene was located on an IncHI2 plasmid (pMLST type ST1), while one isolate (RBL-18-0082-1) carried a truncated version of the *mcr-9* on the chromosome. For two of the seven *E. xiangfangensis* isolates (RPF-17-0249-1, L-1; RBL-18-0082-1, L-2) that carried *mcr-9* on plasmid, there was an internal mutation leading to frameshifts in their respective reading frames (Supplementary Fig. 3). In *E. ludwigii* (RPB-16-0320-1) the *mcr-9* gene was chromosomally located. Regardless of its location, the *mcr-9* was surrounded by IS903-like IS elements described by Kieffer et al.[28] in the upstream region, while IS26-like,

(a)

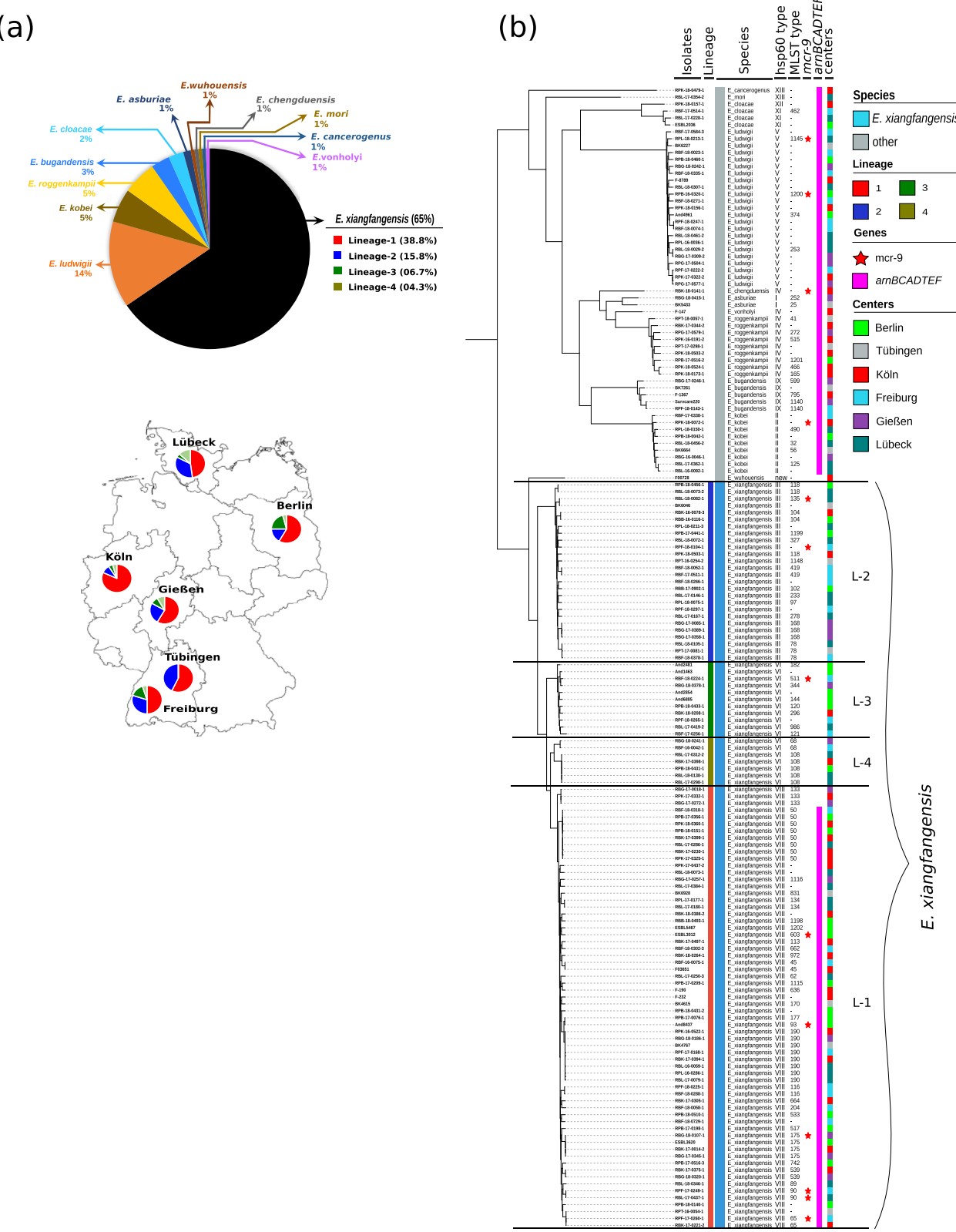

(b)

IS102B, or IS1R IS elements were present downstream of the gene. The *wbuC* gene was present in seven isolates, whereas the *qseB/qseC/*ATPase/*int* structure was detected in only one isolate. Altogether, these results indicate heterogeneity of *mcr-9*-encoding elements.

Studies using the BMD assay showed that colistin resistance in these *Enterobacter* isolates remain unaffected, regardless of whether the *mcr-9* gene was present.

## Colistin resistance and heteroresistance are associated with the presence of *arnBCADTEF*

Comparative genomics of colistin-susceptible versus -resistant isolates showed that resistance patterns associated strongly with the presence of the *arnBCADTEF* gene cassette (*arn*) (Table 1). Isolates of *E. wuhouensis, E. xiangfangensis* lineages L-2, L-3, and L-4, and ST133 isolates of *E. xiangfangensis* L-1 that lacked *arn* showed colistin

**Fig. 1 | Distribution of *Enterobacter* species among six university hospitals in Germany. a** A total of 165 *Enterobacter* isolates were studied. Genome-based species classification showed *E. xiangfangensis* as the most frequent species at all centers. Values in parentheses denote the % of total *Enterobacter* (n = 165) isolates. The outline map of Germany is obtained from https://vemaps.com.
**b** Phylogenomic distribution of the *Enterobacter* isolates obtained between 2011 and 2018 from six different study centers. Whole-genome sequence-based study of 165 *Enterobacter* isolates revealed 65.4% of isolates to be *E. xiangfangensis*. Bayesian Analysis of Population Structure (BAPS) in concordance with the multi-

phylogenomic approach (concatenate core genes (shown here), core- and whole-genome) clustered *E. xiangfangensis* isolates into four lineages. A majority (60% of *E. xiangfangensis* isolates) belonged to Lineage-1, which harbor *arnBCADTEF* gene cassette associated with colistin heteroresistance. Importantly, 9 *E. xiangfangensis* and 4 other *Enterobacter* species isolates carried the recently identified mobile colistin resistance gene *mcr-9*. The *hsp60* types correlated well to the phylogenomic lineages. The lineages were supported by bootstrap values (100). The distribution data for the isolates are provided in the Source data file.

**Table 1 | Colistin resistance and heteroresistance of the *Enterobacter* species**

| *Enterobacter* species (OGRI) | hsp60 cluster | *arnBCADTEF* gene cassette | Colistin resistance/ heteroresistance | Colistin MIC range by BMD (mg/L) | Heteroresistant frequency by PAP (%) | |
|---|---|---|---|---|---|---|
| | | | | | 8 mg/L | 32 mg/L |
| *E. asburie* | I | + | + | 128–256 | 0.44–5.30 | 0.11–1.20 |
| *E. bugandensis* | IX | + | + | 64–>512 | 4.10–25.80 | 1.63–5.28 |
| *E. cancerogenus* | XIII | + | + | 64–128 | 0.75–1.78 | 0.24–2.54 |
| *E. mori* | XIII | + | + | 32–64 | 0.96–1.44 | 0.53–0.55 |
| *E. chengduensis* | IV | + | + | 64–128 | 10.35–14.95 | 6.73–9.05 |
| *E. roggenkampii* | IV | + | + | 64–>512 | 0–9.70 | 0–5.38 |
| *E. vonholyi*[a] | IV | –[a] | –[a] | 2–4 | 0 | 0 |
| *E. cloacae* | XI | + | + | 512–>512 | 0–48.38 | 0.00–24.09 |
| *E. cloacae* | XII | + | + | 512 | 18.69–24.43 | 9.49–12.75 |
| *E. kobei* | II | + | + | 8–>512 | 0–1.19 | 0–0.55 |
| *E. ludwigii* | V | + | + | 4–256 | 0–1.07 | 0–0.14 |
| *E. xiangfangensis* L-I[b] | VIII | +[b] | +[b] | 2–128 | 0–0.75 | 0–0.50 |
| *E. xiangfangensis* L-II | III | – | – | 2–4 | 0 | 0 |
| *E. xiangfangensis* L-III | VI | – | – | 2–4 | 0 | 0 |
| *E. xiangfangensis* L-IV | VI | – | – | 2–4 | 0 | 0 |
| *E. wuhouensis* | New | – | – | 4–6 | 0 | 0 |

A total of 165 isolates of different *Enterobacter* species were studied for their minimum inhibitory concentration (MIC$_{LB}$) and representative isolates of each species were subject to population analysis profile (PAP) test. The MIC$_{LB}$ of individual isolates varied in triplicate and often exhibited the "skipped well" phenomenon suggesting heteroresistance. All the isolates carrying intact *phoPQ/mgrB/arnBCADTEF* genes exhibited MIC$_{LB}$ in the range of 2–>512 mg/L. PAP analysis showed heteroresistance frequency of *Enterobacter* species to vary between none to 48.4%. Different *Enterobacter* species exhibit different range of MIC$_{LB}$ and HRF. Data for individual isolates are presented in Supplementary Data 1b. Key: +: present, –: absent.
[a]A single isolate of *E. xiangfangensis* L-1 (RBL-17-0250-3) and a single isolate of *E. vonholyi* carried truncated *phoQ* gene.
[b]Three isolates of *E. xiangfangensis* (belonging to ST133) lack *arnBCADTEF*.

MIC$_{LB}$ ≤ 4 mg/L (or MIC$_{EUCAST}$ ≤ 2 mg/L). The remaining isolates of *E. xiangfangensis* L-1 and all other species with *arn* exhibited MIC$_{LB}$ > 4 mg/L (or MIC$_{EUCAST}$ > 2 mg/L) (Table 1 and Supplementary Data 1b). Notably, for the *arn* encoding isolates, the MIC$_{LB}$ for colistin varied between 4 and ≥512 mg/L in a species-dependent manner. Isolates of *E. xiangfangensis* (L-1), *E. asburiae*, *E. ludwigii*, *E. bugandensis*, *E. cancerogenus*, *E. chengduensis*, *E. cloacae*, *E. kobei*, and *E. roggenkampii* had MIC$_{LB}$ varying between 2 to ≥512 mg/L. A single isolate of *E. vonholyi*, that encoded *arn*, was observed to be sensitive (due to truncated regulator component *phoQ*; detailed below).

In repeated testing for colistin susceptibility using the BMD method for the same isolate, twofold to fourfold changes in the MIC$_{LB}$ with the "skip wells" phenomenon were observed indicating heteroresistance towards colistin[29]. Using the PAP assay, we calculated heteroresistance frequency (HRF) i.e., the percentage of cells grown on LB agar plate containing colistin as compared to the inoculum used for all the species encountered. Only those isolates that carry an intact *arn/phoPQ* system exhibited heteroresistance, while isolates lacking this system or carrying truncated versions, showed neither the "skip well" phenomenon nor exhibited heteroresistance in PAP assay (Table 1). This data indicated, as with colistin resistance, that an intact *arn/phoPQ* system is required also for colistin heteroresistance.

We examined for the presence of *arn* in *Enterobacter* in additional species that were not detected in this study. Screening of genomes of

*Enterobacter* species retrieved from public databases revealed that, of the 23 *Enterobacter* species known today (July 2022), 19 carry the *arn* gene cassette (Supplementary Data 2). The location of this cassette is always found between the same core genes on the chromosome (Supplementary Fig. 4). In those species lacking *arn*, there is a clean excision of this gene cassette leaving the same flanking genes intact. In summary, all the isolates of *E. xiangfangensis* L-2, L-3, and L-4, *E. hormaechei*, *E. wuhouensis*, and a single isolate (i.e. type strain) of *E. timonensis* lack *arn*. These isolates lacking *arn* form a unified clade (Fig. 1b and Supplementary Fig. 2). As *Enterobacter* species carried *arn* at an identical chromosomal location and were members of a single phylogenomic clade indicated that *arn* was lost during the evolution of *Enterobacter*.

**Heteroresistance levels correlate with the intrinsic level of lipid A L-Ara4N-modification**
To define the nature of structural modifications associated with colistin resistance we extracted lipid A from various isolates of *E. xiangfangensis* grown either in the presence or absence of colistin (2 mg/L) (Fig. 2a and Supplementary Data 4). Only isolates encoding *arn* (most L-1 isolates) could modify their lipid A with L-Ara4N. Such isolates grown in the absence of colistin showed lipid A molecules with only a single L-Ara4N substitution either at the 4′- (R1) or 1′- (R2) phosphate. Growth of these isolates in presence of colistin resulted in a significant overall increase of L-Ara4N modifications, and in part with

(a)

| Species | Isolate | lineage | mcr-9 | arn | MIC | Ara4N-substitution of lipid A species | | | | | |
|---|---|---|---|---|---|---|---|---|---|---|---|
| | | | | | | no colistin | | | colistin (2 mg/L) | | |
| | | | | | | no subst. | 1 Ara | 2 Ara | no subst. | 1 Ara | 2 Ara |
| E. xiangfangensis | RBG-18-0107-1 | I | + | + | 4-16 | 94.2 | 5.8 | 0.0 | 43.1 | 55.1 | 1.8 |
| E. xiangfangensis | RBL-17-0437-1 | I | + | + | 2-8 | 99.6 | 0.4 | 0.0 | 32.6 | 59.9 | 7.5 |
| E. xiangfangensis | RPF-17-0260-1 | I | + | + | 4-8 | 99.8 | 0.2 | 0.0 | 34.5 | 61.4 | 4.1 |
| E. xiangfangensis | BK4615 | I | - | + | 2-16 | 99.9 | 0.1 | 0.0 | 36.3 | 56.2 | 7.6 |
| E. xiangfangensis | RBK-17-0394-1 | I | - | + | 2-32 | 96.7 | 3.3 | 0.0 | 30.1 | 62.4 | 7.5 |
| E. xiangfangensis | RBG-17-0018-1 | I | - | - | 2-4 | 100.0 | 0.0 | 0.0 | 100.0 | 0.0 | 0.0 |
| E. xiangfangensis | RPT-16-0254-2 | II | - | - | 2-4 | 100.0 | 0.0 | 0.0 | 100.0 | 0.0 | 0.0 |
| E. xiangfangensis | RBG-18-0378-1 | III | - | - | 2-4 | 100.0 | 0.0 | 0.0 | 100.0 | 0.0 | 0.0 |
| E. xiangfangensis | RBF-18-0224-1 | III | + | - | 2-4 | 100.0 | 0.0 | 0.0 | 100.0 | 0.0 | 0.0 |
| E. xiangfangensis | RBL-17-0298-1 | IV | - | - | 4-16 | 100.0 | 0.0 | 0.0 | 100.0 | 0.0 | 0.0 |

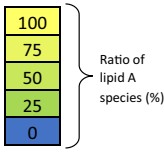

Ratio of lipid A species (%): 100, 75, 50, 25, 0

(b)

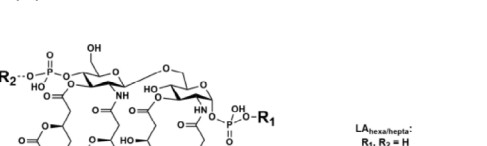

LA$_{hexa/hepta}$:
R$_1$, R$_2$ = H

LA$_{hexa/hepta}$ + 1 Ara4N:
R$_1$= H, R$_2$ = Ara4N or R$_1$= Ara4N, R$_2$ = H

LA$_{hexa/hepta}$ + 2 Ara4N:
R$_1$, R$_2$ = Ara4N

Ara4N =

(c)

| Species | Isolate | arn | phoPQ | Ara4N-substitution of lipid A species | | |
|---|---|---|---|---|---|---|
| | | | | no colistin | | |
| | | | | no subst. | 1 Ara | 2 Ara |
| E. roggenkampii | RPB-17-0516-2 | + | + | 83.9 | 16.1 | 0.0 |
| E. ludwigii | F-8789 | + | + | 93.9 | 6.1 | 0.0 |
| E. bugandensis | RBG-17-0246-1 | + | + | 58.1 | 40.1 | 1.8 |
| E. cloacae | RPK-18-0157-1 | + | + | 70.6 | 28.0 | 1.4 |
| E. chengduensis | RBK-18-0141-1 | + | + | 62.5 | 36.1 | 1.4 |
| E. cancerogenus | RPK-18-0479-1 | + | + | 62.1 | 34.1 | 3.8 |
| E. mori | RBL-17-0354-2 | + | + | 94.3 | 5.7 | 0.0 |
| E. kobei | BK6664 | + | + | 93.5 | 6.5 | 0.0 |
| E. asburiae | BK5433 | + | + | 76.9 | 22.7 | 0.4 |
| E. vonholyi | F-147 | + | truncated | 96.8 | 3.2 | 0.0 |
| E. wuhouensis | F00728 | - | + | 100.0 | 0.0 | 0.0 |

(d)

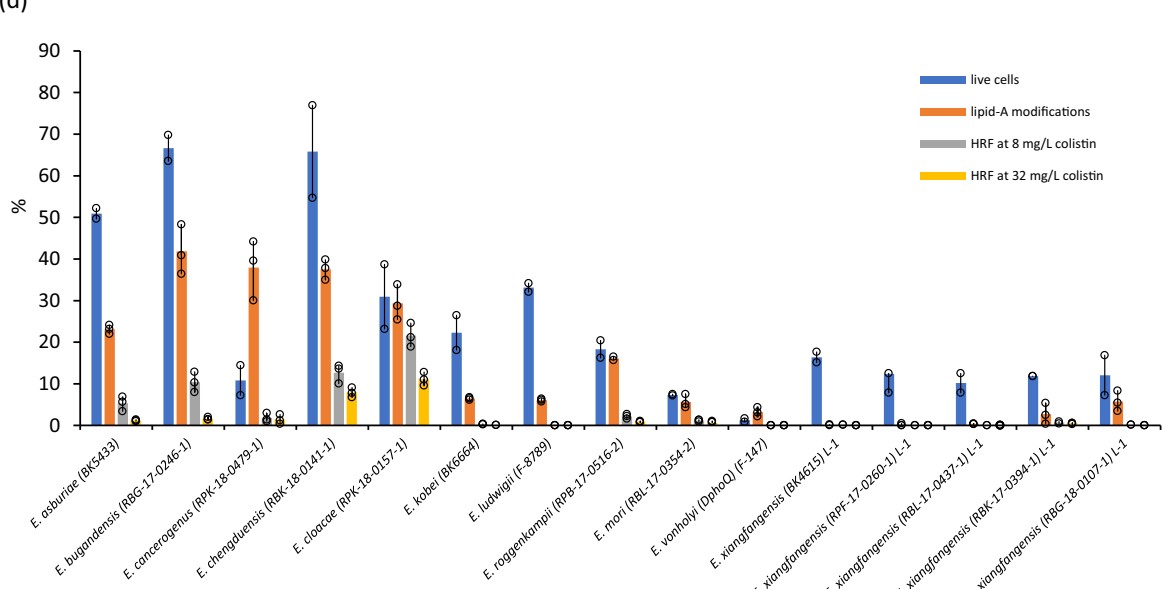

double substitution. Strains that lacked the *arn* gene cassette did not show any modifications of lipid A. The general structure of *Enterobacter* lipid A is shown in Fig. 2b, all lipid A species observed in our MS analyses are listed in Supplementary Data 4. Representative MS spectra for *E. xiangfangensis* isolates are shown in Supplementary Fig. 5 and such spectra are the source of reported ion intensities used for lipid A ratio calculation (Supplementary Data 4). Of note, in isolates carrying the *mcr-9* gene, no phosphoethanolamine (PEtN) modification was observed, indicating that *mcr-9* is either non-functional or not induced under the growth conditions used. The same MS-based lipid A analysis was performed for representative strains of the remaining 11 *Enterobacter* species grown in colistin-free media. We detected varying levels of L-Ara4N modification of lipid A in individual isolates representing different species (Fig. 2c).

**Fig. 2 | Analysis of L-Ara4N-modification levels of lipid A and their correlation with heteroresistance frequency and the number of cells that survived after colistin treatment. a** Heat map summarizing the MS-based analysis of L-Ara4N-modification levels of lipid A in selected *E. xiangfangensis* strains. Corresponding, representative mass spectra are shown in Supplementary Fig. 5, further details are specified in Supplementary Data 4. **b** General chemical structure of *Enterobacter* lipid A. **c** Heat map summarizing the MS-based analysis of L-Ara4N-modification levels of lipid A in 11 other *Enterobacter* species observed in this study. **d** Various *Enterobacter* isolates grown at identical conditions (LB broth, O.D.$_{600}$ 0.8–1.0, without colistin) were analyzed for L-Ara4N-modification of lipid A ((see panels **a** and **c**), Supplementary Data 4) and for their correlation with % of heteroresistant cells (on a plate containing 8 and 32 mg/L of colistin), and % of cells survived after colistin treatment (64 mg/L for 15 min). The % of heteroresistance CFUs at 8 and

32 mg/L was positively associated with L-Ara4N-modification levels of lipid A (Pearson's correlation coefficient, $r = 0.88$ and 0.73, respectively). This data suggested colistin heteroresistance to be directly associated with such levels. Given that only a fraction of isogenic growing population survives on plates containing 8 and 32 mg/L of colistin, we determine the % of cells that can survive after colistin treatment (64 mg/L for 15 min) by Live/Dead™ staining assay. This assay showed a positive association ($r = 0.86$) between % of cells survived and L-Ara4N-modified lipid A. This data indicated that only a certain number of cells from isogenic growing population carried modified lipid A and can survive when exposed to colistin, resulting in the heteroresistance phenomenon. This data show that cells with L-Ara4N-modified lipid A form the founder population of the colistin heteroresistance (also see Supplementary Table 5). Source data are provided as a Source data file.

We wondered if the levels of lipid A modification seen in cultures grown in the absence of colistin reflected the fraction of hetero-resistant bacteria observed by the PAP assay. In addition, we compared HRFs to the levels of lipid A modified with L-Ara4N for representative strains of all the species studied when grown under the same conditions. We found a very strong positive correlation between the proportions of L-Ara4N-modified lipid A and their HRF (Pearson correlation coefficient: 0.88 and 0.73 for 8 and 32 mg/L colistin, respectively) indicating heteroresistance essentially reflects the level of intrinsically modified lipid A when isolates are grown in media without colistin (Fig. 2d and Supplementary Table 5).

We next determined if L-Ara4N-modification levels of lipid A represented minor sub-populations within isogenic growing population. To differentiate between intact and damaged membranes from isogenic growing population, we used LIVE/DEAD™ staining following exposure of bacteria to a high concentration (64 mg/L) of colistin for 15 min and observed cell populations using fluorescence microscopy (Supplementary Fig. 6). Analysis of *arn* carrying *Enterobacter* species revealed sub-populations of cells with intact membranes, inferring their survival during colistin treatment. The percentage of cells survived showed positive correlation ($r = 0.86$) with the levels of modified lipid A. Exposure of bacteria for longer time (up to 4 hours) at high colistin concentration (64 mg/L) did not result in additional killing and the percentage of cells that survived were comparable to that of 15 min. The high correlation of lipid A L-Ara4N-substitution levels to HRF and proportions of live cells under identical growth conditions suggests that a genetic basis underlies the presence of sub-populations observed for the different species.

**Allelic differences in *phoQ* and *mgrB* determine colistin resistance and heteroresistance**

In *Enterobacterales*, expression of *arn* has been shown to be dependent on *pmrAB, phoPQ*, and the PhoPQ-feedback inhibitor *mgrB*[22,30]. To determine the role of genetic components involved in the colistin resistance or heteroresistance, we created and studied isogenic mutants of *arn* (i.e., *arnBCADTEF* gene cassette), *pmrAB, phoPQ*, and *mgrB* in a representative *E. xiangfangensis* L-1 isolate RBK-17-0394-1 (henceforth called Ex394).

While the parental isolate Ex394 exhibited a MIC$_{LB}$ of 2–32 mg/L towards colistin, the Δ*arn* mutant had a MIC$_{LB}$ of only 0.5 mg/L (Fig. 3). Reintroduction of the *arn* on the low copy pBBR1-MCS2 plasmid reinstated the MIC$_{LB}$ to 8-16 mg/L, demonstrating an absolute requirement of this gene cassette for colistin resistance activity. Deletion of *phoPQ* also lowered colistin resistance (2 mg/L), while the MIC$_{LB}$ exhibited by the Δ*pmrAB* mutant was comparable to that of the parental strain at 4–16 mg/L. For the Δ*mgrB* mutant, encoding the negative regulator of PhoPQ, high MIC$_{LB}$ levels of 128–>512 mg/L were observed. Genetic complementation by reintroduction of the respective gene deleted in these mutants restored MIC$_{LB}$ levels to that seen with the parental Ex394 strain.

We extended this data by examining the levels of L-Ara4N substitution of lipid A. Lipid A preparations from the Δ*arn* and Δ*phoPQ* mutants were totally devoid of L-Ara4N modifications, while the Δ*pmrAB* and the parental strain Ex394 showed similar levels of lipid A substitutions under both growth conditions (Supplementary Data 4). The Δ*mgrB* mutant exhibited high levels of L-Ara4N substitution even when grown in the absence of colistin. In the LIVE/DEAD™ staining assay the Δ*mgrB* mutant showed higher numbers of viable bacteria as compared to WT, while Δ*arn* and Δ*phoPQ* had relatively lower numbers of surviving bacteria following colistin treatment (Supplementary Fig. 6 and 7).

To understand variations in the MIC$_{LB}$ and HRFs of different *Enterobacter* species we examined *phoPQ/mgrB* loci for allelic differences. Analysis of nucleotide sequences of gene and their promoter regions showed high conservation for the *mgrB* locus (Supplementary Data 6). Significant differences were noted within the coding regions of PhoP and PhoQ, while the respective *phoPQ* promoter region was highly conserved. Species-wise differences were observed in the amino acid sequences of PhoPQ suggesting allelic changes could lead to differences in resistance and heteroresistance levels.

As the various *Enterobacter* species that harbor different alleles of PhoPQ have a wide range of MIC$_{LB}$ and HRF, we introduced the *phoPQ* alleles from *E. bugandensis* RBG-17-0246-1 (Eb246) and *E. roggenkampii* RPB-17-0516-2 (Er516), that each exhibit MIC$_{LB}$ of between 64 and >128 mg/L, into the Ex394Δ*phoPQ* mutant (Fig. 4). Complementation with the heterologous *phoPQ$_{246}$* or *phoPQ$_{516}$* genes in Δ*phoPQ$_{394}$* mutant resulted in recombinants that exhibited MIC$_{LB}$ of 64–128 mg/L, respectively. The HRF increased to 42.33 ± 4.78% and 10.83 ± 1.43% on 8 and 32 mg/L colistin with *phoPQ$_{246}$* complementation. For the strain complemented with *phoPQ$_{516}$*, HRFs were 7.13 ± 2.75% and 1.05 ± 0.19% on 8 and 32 mg/L colistin, respectively (Supplementary Table 6).

These results were also examined by analysis of transcript levels of the *arnBCADTEF* operon. Compared to complementation with *phoPQ$_{394}$*, the transcript levels of *arnBCADTEF* increased by 20.84-fold when complemented with *phoPQ$_{246}$* and 2.75-fold with the *phoPQ$_{516}$* allele following transfer into the Ex394Δ*phoPQ* mutant (Supplementary Fig. 8). Mass spectrometric analysis of L-Ara4N modifications of lipid A indicated that it was highest in Δ*phoPQ* strain complemented with *phoPQ$_{246}$* followed by *phoPQ$_{516}$* and then *phoPQ$_{394}$* (Supplementary Table 5). This was further validated using the LIVE/DEAD assay (Supplementary Fig. 6 and 7). We observed higher ratios of viable bacteria for the *phoPQ$_{246}$* complemented strain, followed by *phoPQ$_{516}$* and the *phoPQ$_{394}$* complemented strains.

To differentiate between the individual role of alleles of *phoP* and *phoQ*, we created hybrid versions of *phoPQ* by generating chimeric combinations of *phoP* and *phoQ* from each of Ex394, Eb246, and Er516, and transferring them to the Δ*phoPQ$_{394}$* mutant (Fig. 4). Among the different combinations of *phoP* and *phoQ* alleles, those that carried *phoQ$_{246}$* had the highest MIC$_{LB}$ and HRF levels, followed by *phoQ$_{516}$*, and was lowest with *phoQ$_{394}$*. Unlike the results obtained with the

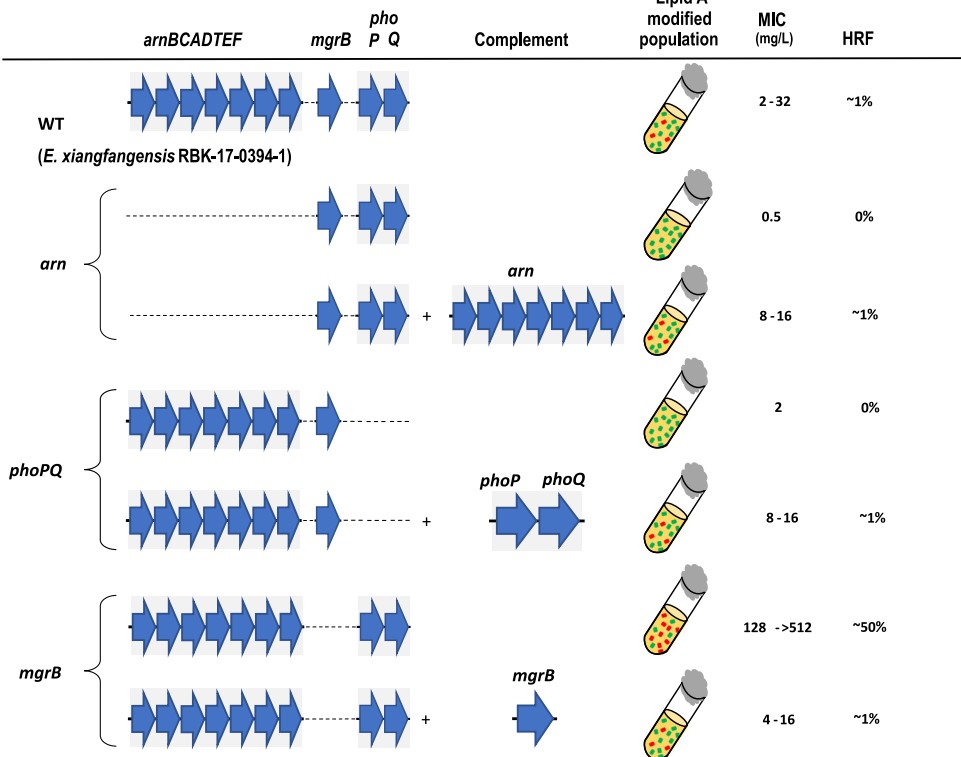

**Fig. 3 | Genetic determinants of heteroresistance in *Enterobacter* species.** Comparative genomics of colistin-resistant and -sensitive isolates revealed that *arnBCADTEF* is necessary for colistin heteroresistance. In *Enterobacterales*, *arnBCADTEF* expression is controlled by the TCS PhoPQ, while the activity of the PhoPQ pathway is inhibited by the small transmembrane protein MgrB. Deletion mutants were created in *E. xiangfangensis* L-1 isolate RBK-17-0394-1 (WT) and studied for the MIC$_{LB}$ and HRF (number of cells recovered on LB agar plate containing 8 or 32 mg/L of colistin/number of cells inoculated). MIC$_{LB}$ was measured by BMD while HRF was calculated by PAP test. Analysis of mutants showed the *phoPQ/arnBCADTEF* axis is absolutely required for the colistin heteroresistance, while MgrB acts as a negative regulator of *arnBCADTEF* resulting in colistin susceptibility. As reported previously by Kang et al.[22] for *E. cloacae*, PmrAB did not contribute to colistin resistance in *E. xiangfangensis*. Source data are provided as a Source data file. The MIC data is provided in the Source data file.

respective *phoQ* alleles, no significant difference in MIC$_{LB}$ was observed with *phoP* alleles deriving from the 246/516/394 strains. These data implicate the PhoQ sensor component of the PhoPQ two-component system as a major determinant in the activation of *arnBCADTEF* transcription and expression.

We next addressed the contribution of the *mgrB* gene from Eb246/Er516, whose sequences are identical, through the introduction of the complementing plasmids into the Δ*mgrB$_{394}$* mutant. Recombinants complemented with the *mgrB$_{246/516}$* allele exhibited an elevated MIC$_{LB}$ of 64 mg/L, with a HRF value of 2.10 ± 1.13% and 0.96 ± 0.66% at 8 and 32 mg/L of colistin, respectively. In contrast, complementation with the homologous *mgrB$_{394}$* allele reduced colistin MIC$_{LB}$ values to 4-16 mg/L and expressed a HRF value of 0.34 ± 0.24% and 0.22 ± 0.09% at 8 and 32 mg/L of colistin, respectively (Fig. 4).

**An assay for unambiguous determination of colistin resistance in *Enterobacter* spp.**

For the species *Salmonella typhimurium* and *Escherichia coli*, the TCS PhoPQ senses acidic pH as an environmental signal and regulates components required for the adaptation and survival of bacteria under these conditions. We reasoned that screening of *Enterobacter* isolates grown in culture media at a low pH would overcome MgrB-mediated suppression of PhoQ and lead to constitutive expression of the *arn* gene cassette, thereby promoting colistin resistance. We examined all the isolates in this study for colistin resistance by growing cultures at pH 7 and pH 5. Those *Enterobacter* isolates that carry intact copies of genes corresponding to the *phoPQ/mgrB/arn* axis, showed 4 to 16-fold higher MIC$_{LB}$ when tested at pH 5 and generally exceeded 256 mg/L (Fig. 5a and Supplementary Data 1). The PAP test carried out by

growing Ex394 on colistin LB agar plate adjusted for pH 5 showed a higher HRF (35.41 ± 7.78% and 5.33 ± 0.29%, on 8 and 32 mg/L colistin, respectively) than that at pH 7 (0.745 ± 0.27% and 0.505 ± 0.15%) (Fig. 5b). Mass spectrometric analysis of Ex394 grown at pH 5 and pH 7 validated these results and showed higher L-Ara4N-modification levels of lipid A in isolates grown at pH 5 as compared to pH 7 (Supplementary Data 4). Thus, at pH 5 suppression of PhoQ by MgrB is overcome, and results in the activation of the PhoPQ-dependent regulon that includes the *arnBCADTEF* gene cassette.

## Discussion

Taxonomic uncertainty associated with identifying members of the ECC together with problems relating to routine microbiology susceptibility testing for colistin resistance underestimate both the true prevalence of these bacteria in causing infections and mask the extent of how widespread this resistance is in these bacteria. ECC isolates, along with *E. coli* and *Klebsiella pneumoniae*, often harbor carbapenemases, forcing the therapeutic use of colistin worldwide. An expansion of genomic lineages of colistin-resistant *Enterobacter* species would be potentially concerning as relatively few antibiotic combinations remain as further treatment options. Adding to these concerns, problems associated with heteroresistance pose a serious threat because infections with *Enterobacter* are notoriously persistent and difficult to treat. The development of simple, reliable diagnostic methods that recognize resistance and heteroresistance is therefore urgently needed.

Genome-based taxonomy provided consistent and robust topologies allowing the identification of *E. xiangfangensis* as the predominant clinical species both in Germany as well as worldwide.

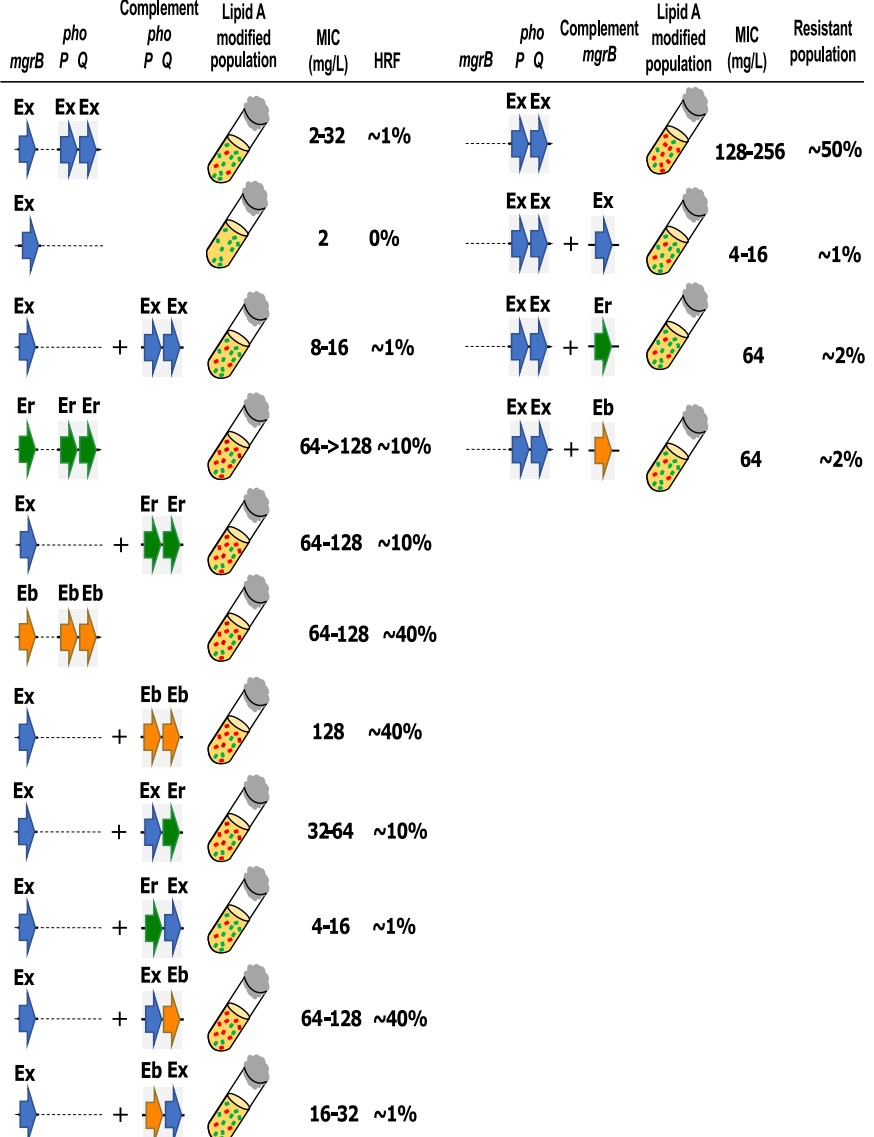

**Fig. 4 | Heteroresistance capabilities of *phoP-phoQ* hybrid and *mgrB* complemented mutants.** The hybrid complement mutants were created by combining *phoP* and *phoQ* genes either of *E. xiangfangensis* RBK-17-0394-1 (Ex), *E. bugandensis* RBG-17-0246-1 (Eb), and *E. roggenkampii* RPB-17-0516-2 (Er). Plasmids harboring *phoP-phoQ* hybrids were transferred to Ex394Δ*phoPQ*. Complementation of the *phoQ* from the isolates exhibiting higher colistin heteroresistance i.e., Eb and Er to the Δ*phoPQ* mutant of *E. xiangfangensis* RBK-17-0394-1 resulted in the colistin resistance equivalent to respective parent strains. Complementation of the Δ*mgrB* mutant of *E. xiangfangensis* RBK-17-0394-1 with the *mgrB* from Eb (which has an identical amino acid sequence as of Er) resulted in slightly lesser inhibition due to changes in the binding energy (detailed in Supplementary Fig. 9). The MIC data is provided in the Source Data file.

To improve the utility of this taxonomic scheme and further encourage studies on historical isolates, we carried out a backward compatibility analysis to the previously developed based on *hsp60* gene analysis[31]. This enabled retrospective analysis of *Enterobacter* isolates from several studies published previously, e.g. Hoffman et al. (comprising isolates from both the USA and European countries)[31], Paauw et al. (The Netherlands)[7], Morand et al. (France)[32], Garinet et al. (France)[32], Moradigaravand et al. (UK)[33], Akbari et al. (Iran)[34], Peirano et al. (carbapenemase-resistant worldwide isolates)[12], Wang et al. and Zhou et al. (China)[35,36]. From these retrospective studies we found a predominance of *hsp60* clusters -III, -VI, and particularly -VIII, which are essentially identical to lineage-1 isolates of *E. xiangfangensis* determined here, suggesting that this is the most commonly occurring species among clinical isolates worldwide, going back over a period of at least two decades (Supplementary Data 2).

Although heteroresistance towards colistin in *Enterobacter* species was first reported in 2007[37], the often inconsistent MIC values and complications in interpreting data from the "skipped well" phenomenon have made determinations on the overall prevalence of colistin resistance difficult[20,22,38,39]. Thus, according to current WHO classifications, *Enterobacter* species are listed as being generally susceptible to colistin[15,16]. The emergence of transferable colistin resistance through plasmids, and in particular the extremely high occurrence of *mcr-9* (~20%) in *Enterobacter* spp. had added further urgency in understanding the true levels of colistin resistance in ECC[40].

To correlate lipid A profiles, heteroresistance frequency and data from microscopic studies, we determined MICs using identical growth conditions i.e., using LB broth. When compared to MIC levels obtained with standard EUCAST protocols, the MIC$_{LB}$ values were either similar or two-fold higher (Supplementary Data 1b). As with the EUCAST approach the "skipped well" phenomenon was also observed during MIC$_{LB}$ determinations. Levels of colistin resistance depend on the extent of modified lipid A molecules with substitutions that reduce the electronegative potential of lipopolysaccharide (LPS). From this study

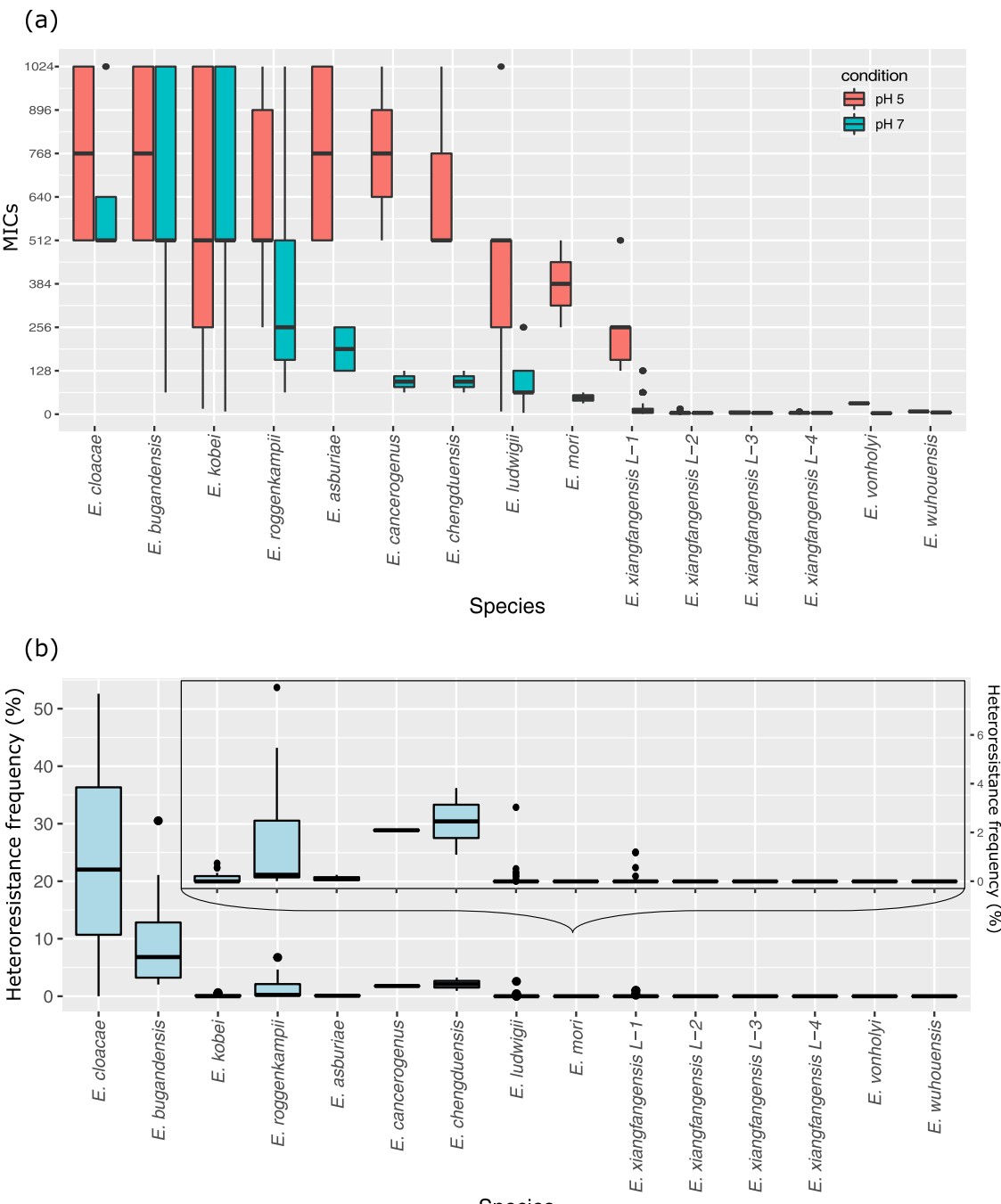

**Fig. 5 | Minimum inhibitory concentration (MIC$_{LB}$) and heteroresistance frequency (HRF) of different *Enterobacter* species. a** MIC$_{LB}$ and **b** HRF. The MIC$_{LB}$ and HRF were studied by BMD assay and PAP test for isolates of 12 species observed in this study. The red and blue boxes (in **a**) depict MIC$_{LB}$ at pH 5 and 7, respectively. *Enterobacter* species exhibiting lower HRFs are shown also in the inset (in **b**). The MICs and HRFs for each individual isolate are provided in the Source data file. Keys for box and whisker plot: The black center line denotes the median value (50th percentile), while the red or blue box contains the 25th to 75th percentiles of dataset. The black whiskers mark the inter-quartile range (1.5× IQR) and outlier points are shown individually.

we conclude that L-Ara4N modification of lipid A is solely responsible for colistin resistance (and heteroresistance) in ECC, as we observed PEtN-based substitutions only in up to 1.2% (Supplementary Data 4), even in those isolates that carry *mcr-9* but lack the *arnBCADTEF* gene cassette. Based on the taxonomic scheme developed and from functional characterization of lipid A modifications in representative isolates we conclude that 20 of 23 *Enterobacter* species are inherently colistin resistant. Interestingly, for the remaining species, where most isolates are members of the sublineages L-2, L-3, and L-4 of *E. xiangfangensis*, we show clonal loss of the *arnBCADTEF* gene cassette

associated with a clean excision event. As these represent a significant subset of *Enterobacter* isolates found at different sites in this study, the implication of this observation, i.e., if it is adaptive or provides growth advantages needs to be explored. Alternatively, the ubiquitous presence of this gene cassette in several members of the order *Enterobacterales* including *Enterobacter*, *E. coli*, and *K. pneumoniae* suggest that, in these lineages, it could be rapidly reacquired from close relatives.

The species-specific heteroresistance observed in this study and previous *hsp60* cluster-based studies[20,26] showed that the degree of

heteroresistance differs across *Enterobacter* species (Table 1). In *Enterobacterales*, resistance towards colistin is complex and multifactorial requiring genes encoding structural components as well as those involved in LPS transport and its modification, together with associated regulatory genes[41]. Quantitative lipid A analysis of isogenic mutants and complementation analysis demonstrated that colistin resistance requires the presence of an intact *arn/phoPQ/mgrB* axis. Strikingly, when complemented, recombinants also exhibited skip-well phenomenon in $MIC_{LB}$ testing assays, indicating that these genes also control inherent heteroresistance capacity[29,42].

A major finding in this study using heterologous complementation assays was that those allelic differences in the two component *phoPQ* sensor-activator system from different species govern levels of heteroresistance. Thus, isolates that exhibit high heteroresistance levels could confer this property, when the respective *phoPQ* alleles were transferred to an isogenic mutant strain from an isolate exhibiting low heteroresistance propensity. Quantitative transcriptional analysis demonstrated overexpression of the *arnBCADTEF* operon in the isogenic mutant depending on the *phoPQ* allele introduced. This increased transcription also corresponded to an increase in the level of L-Ara4N modification of lipid A in the complemented recombinant strains. By creating chimeric *phoPQ* allelic pairs, we demonstrated that qualitative changes in the periplasmic membrane interacting domain of the sensor PhoQ control expression levels of the *arnBCADTEF* gene cassette. Using a similar approach, we determined that the level of colistin resistance is also determined by allelic differences in its inhibitor peptide MgrB.

Our observations are supported by molecular modeling studies. The PhoQ-MgrB interactions predicted that introduction of the altered $mgrB_{246/516}$ allele from a highly resistant strain would significantly alter the $MIC_{LB}$ level of the moderately colistin resistant strain Ex394 because of the higher activation energy required for its interaction with the $phoPQ_{394}$ allele (Supplementary Fig. 9). As predicted, the introduction of $mgrB_{246/516}$ into Ex394$\Delta mgrB$ enhanced its $MIC_{LB}$ suggesting less efficient repression of the endogenous PhoQ sensor component by this variant. Detailed analysis of PhoQ-MgrB indicated that the α/β-core of PhoQ (aa 43–180) interacts with the C-terminal aa residues 34–47 of MgrB. Our data extends data from a recent study showing the importance of the interaction of the C-terminal periplasmic region of MgrB with PhoQ.

Previous studies showed that two pathways leading to acquired colistin resistance may exist in various *Enterobacter* and *Klebsiella* species, one involving the PmrAB, and another associated with PhoPQ[41]. Our data are consistent with the role of the TCS PhoPQ and its inhibitor MgrB, but do not support a role for the PmrAB pathway. This is consistent with data presented by Kang et al.[22] for *E. cloacae*, who did not find any role of *pmrAB* in colistin resistance. It was previously shown that colistin-resistant sub-populations of an isolate of *E. cloacae* increased in peritoneally infected mice even in the absence of co-treatment with colistin, and that this property was associated with the phagocytic capacity of resident macrophages[38]. This selective increase in the frequencies of these sub-populations is attributed to components of the innate immune response including cationic antimicrobial peptides, reactive oxygen species generated by NADPH oxidases and lysozyme[38].

We show here that these sub-populations are already present during growth in broth cultures without antibiotics suggesting that a fraction of an isogenic population of bacteria experience microenvironment changes that results in metabolic heterogeneity[43–45]. These includes changes in osmolarity, oxygen tension, ion flux, pH, as well as periplasmic transmembrane proteins that are all monitored by periplasmic histidine kinase sensors, such as PhoQ. In the case of *S. typhimurium* and other related bacteria, activation of the PhoQ sensor by antimicrobial peptides and acidic pH are separate and additive[46,47]. Our studies suggest a pre-eminent role for environmental pH as a

signal since cells pre-grown at acidic pH exhibited higher $MIC_{LB}$ and HRFs, which is further elevated when cells are pre-grown at acidic pH and in presence of colistin. Under conditions of growth at physiological pH, heteroresistance is a function of the qualitative differences in protein interactions that govern PhoPQ-MgrB interactions in subpopulations sensing microenvironmental pH differences. Flooding the growth environment with protons would constitutively activate PhoPQ and as expected, prior exposure of bacterial cells in the absence of colistin at pH 5 induced high levels of lipid A L-Ara4N-modification and exhibited elevated levels of HRF as compared to those at pH 7 (Fig. 6). Both acidic pH and antimicrobial peptides are likely to be encountered in the host viz., during the passage of the gastrointestinal tract or within the phagolysosomes of macrophages following phagocytosis and sensing of these signals would activate transitory gene expression programs to enable bacterial resilience in responding to the varying environments encountered during infection.

Our data establish a comprehensive overview of the major *Enterobacter* species that are associated with BSI worldwide. The generation of species-specific profiles—using MALDI-TOF, based on the taxonomic scheme described here—is highly desirable and would provide for rapid and robust species designations in clinical, microbiological laboratories. We show that members of the species *Enterobacter* are inherently colistin-resistant due to L-Ara4N modification of lipid A. Finally, we identify environmental fluctuations in pH as the major determinant of heteroresistance levels via sensing by the PhoPQ two-component regulatory system and its inhibitory MgrB peptide and exploit this information to develop a simple, robust assay that unambiguously monitors colistin resistance in *Enterobacter* spp. and would avoid treatment failures in healthcare institutions worldwide. The elements driving clonal heteroresistance are also highly conserved in the order *Enterobacterales*, such as in the species *Klebsiella*, thereby opening the path for generic tests to accurately determine colistin resistance across a broad spectrum of clinically relevant bacteria.

## Methods

### Ethics
The study was approved by the ethics committee of the coordinating site (Cologne) and all sampling sites (Ethical committee approval, coordinating site Cologne: EA4/018/14).

### Isolates and antibiotic resistance profiles
Bacterial strains used in this study are listed in Supplementary Data 1. Cultures were grown in Luria-Bertani (LB) broth (per liter composition: 10 g tryptone, 5 g yeast extract, 10 g NaCl, pH 7) (BD Difco™ Dehydrated Culture Media).

A total of 480 clinical *Enterobacter* isolates (bloodstream, $n = 408$ and colonization $n = 72$) were obtained between 2016 and 2018 from six university hospital centers (Berlin, Lübeck, Cologne, Gießen, Tübingen, and Freiburg) within a surveillance project targeting multidrug-resistant bacterial organisms (R-Net: https://www.dzif.de/en/node/915) (Supplementary Data 1a). Bloodstream infection isolates were obtained through routine diagnostics following clinical indication. Colonization isolates were obtained from newly admitted patients (<48 h hospitalization) on selective agar plates for resistant isolates. Of 480 isolates, 141 were randomly selected (~one-third from each site) and supplemented with 24 historical isolates circulating in Germany (between year 2011-2015, random selection). A total of 165 isolates (bloodstream infections ($n = 96$) and rectal swabs ($n = 69$)) were sequenced and studied (Supplementary Data 1b). These isolates were identified as *Enterobacter* spp. using MALDI-TOF (VITEK MS) or the VITEK2® GN ID card (bioMérieux, France). Antibiotic resistance phenotypes were determined using the VITEK2® system.

Colistin resistance was determined by calculating minimum inhibitory concentrations (MIC) using the broth microdilution (BMD) using two methods (i)- following CLSI-EUCAST Polymyxin Breakpoints

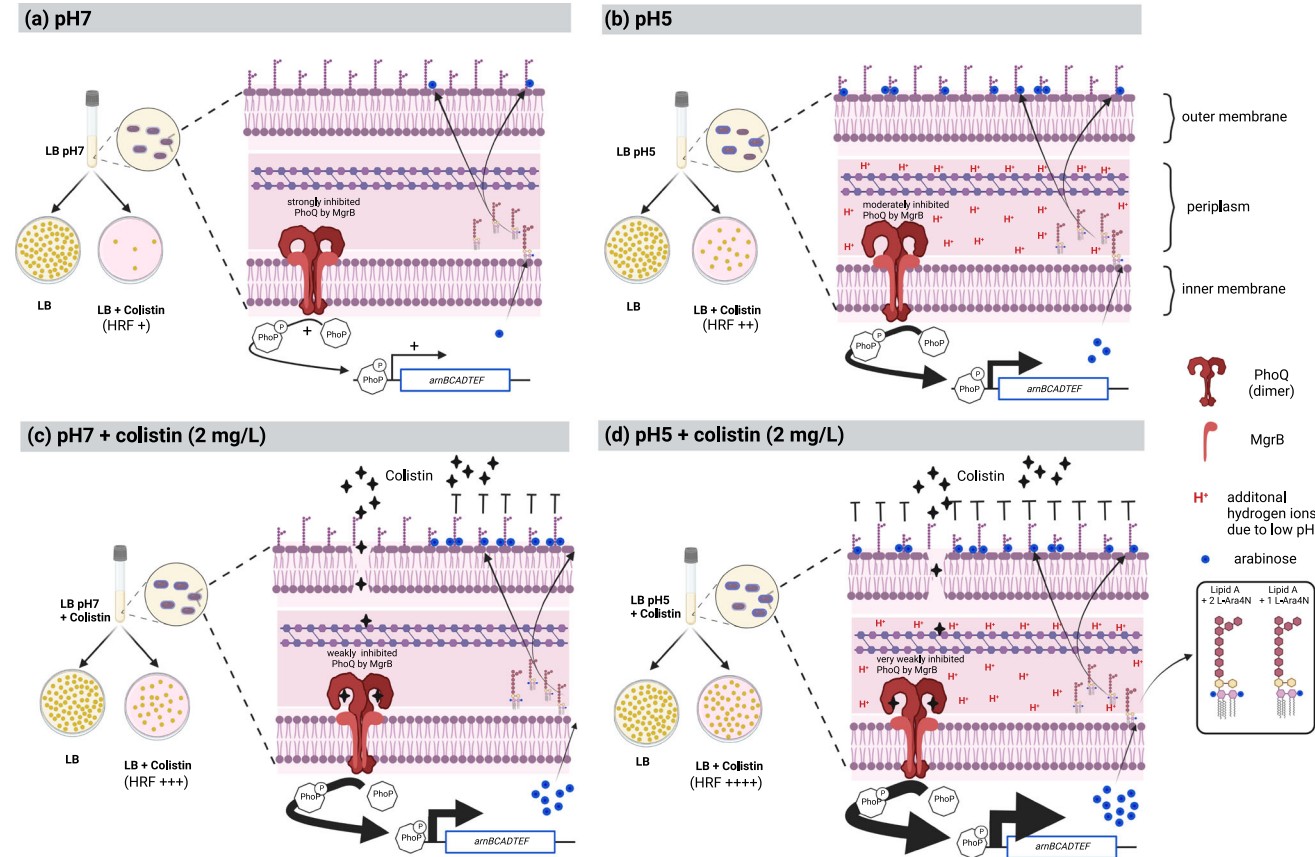

**Fig. 6 | Model of mechanistic pathways leading to colistin heteroresistance.** The PhoQ component (of the two component system PhoPQ) senses environmental conditions (for e.g., low pH and cationic antimicrobial peptide (CAMP)), which results in a change in its conformation leading to phosphorylation of PhoP. Phosphorylated PhoP binds to the promoter of many genes, including *arnBCADTEF*. Expression of *arnBCADTEF* results in the addition of L-Ara4N to lipid A, which changes the overall charge of the outer membrane causing colistin resistance. **a** Under normal growth conditions (LB, pH 7) PhoQ of the overall growing population is strongly inhibited by MgrB[30]. As a result of microenvironmental changes experienced during growth, a portion of the isogenic growing population has activated PhoQ with concomitant expression of the *arnBCADTEF* loci that result in L-Ara4N-modified lipid A. Only these cells survive when exposed to colistin (HRF + ). Overall, levels of modified lipid A are low. **b** As a result of low pH (pH 5) during growth conditions, MgrB is prevented from interacting with PhoQ, resulting in higher expression of the *arnBCADTEF* cassette and subsequent L-Ara4N-modified lipid A in a higher number of cells. Quantitative lipid A analysis confirms these observations and these populations exhibit increased survival (HRF++). **c** In the presence of subinhibitory concentrations of a cationic antimicrobial peptide such as colistin (2 mg/L), the antibiotic penetrates the outer membrane and is sensed by PhoQ strongly weakening interactions with MgrB and thereby releasing expression of *arnBCADTEF*. This results in L-Ara4N-modified lipid A in a higher number of bacteria. When exposed to colistin, these populations show relatively high survival levels (HRF+++). **d** As a result of low pH (pH 5) and antimicrobial peptide colistin during growth, PhoQ experiences a cumulative effect, and thus MgrB inhibition is strongly prevented, resulting in heightened expression of *arnBCADTEF* cassette and subsequent arabinose modified lipid A in a higher number of cells. These populations exhibit increased survival (HRF++++). (Core sugars and fatty acids of the LPS are only drawn symbolically and do not reflect exact structures).

Working Group recommendations[48] and (ii) using LB medium with higher inoculum. In brief, 10 μL from a 30% glycerol stock (in LB broth) was revived in 2 mL LB broth (in pre-sterilized 14 mL PP tube, Greiner Bio-One, Cat. No. 187261) by incubation at 37 °C for 18 h at 180 rpm. The culture was diluted 1:100 in fresh 5 mL Mueller Hinton Broth 2 cation adjusted (MHB2) broth (Merck Millipore, Cat. No. 90922), in 14 mL PP tube and incubated at 37 °C at 180 rpm till an O.D.600 of 0.08–0.1 (equivalent to 0.5 McFarland standard) or if needed, adjusted by addition of further MHB2 broth[49]. For BMD, a 100 μL inoculum was added to 100 μL of MHB2 pre-supplemented with colistin sulfate to obtain a range of 512, 256, 128, 64, 32, 16, 8, 4, 2, 1, 0.5, and 0 mg/L in 96-well plate using a pipetting robot (Assist Plus, Integra). Plates were incubated without shaking at 37 °C for 16–20 h, and readings were taken by a Tecan 96-well plate reader (Infinite M200 Pro, Tecan Group Ltd. Maennedorf, Switzerland) at 600 nm. *E. coli* ATCC 25922 was used as a control for MICs. To correlate the results from other experiments (microscopy, PAP test and lipid A levels), we also determined the MICs in LB broth (MIC_LB). In all, 10 μL inoculum from the culture grown to O.D.600 of 0.8–1.0 was added to 190 μL of LB broth pre-supplemented

with colistin sulfate. The pH experiments were carried out in both MHB2 and LB broth. The pH of the broths were adjusted using 1 M HCl or NaOH. MIC experiments were repeated three times. To compare the MICs at identical growth conditions for other experiments description of MIC_LB is preferred across the manuscript.

## Whole-genome sequencing, genome-based taxonomy, and comparative genomic analysis

For whole-genome sequencing, bacterial DNA was isolated from overnight cultures using the PureLink Genomic DNA isolation kit (Thermo Fisher Scientific, Germany) and libraries were prepared using the Nextera XT kit (Illumina), individually tagged libraries and sequenced (2 × 300 bp/2 × 150 bp) on the MiSeq/NextSeq platform (Illumina)[50]. For ten isolates harboring the *mcr-9* gene, long-read sequences were obtained. The library was prepared using the native barcoding kit (EXP-NBD103, Oxford Nanopore Technologies) and 1D chemistry (SQK-LSK108, Oxford Nanopore Technologies). Sequencing was performed on a MinION sequencer (Oxford Nanopore Technologies) using a SpotON Mk I R9 Version flow Cell (FLO-MIN106, Oxford

Nanopore Technologies). Genome sequences are deposited in the public database under the BioProject accession number PRJNA622426.

For bioinformatic analysis, default parameters were used for all software unless otherwise specified. Illumina raw reads were filtered using Trimmomatic v0.36. ONT long reads demultiplexing and adapter trimming were performed using Porechop v0.2.4 (https://github.com/rrwick/Porechop). The reads were examined by FastQC (https://www.bioinformatics.babraham.ac.uk/projects/fastqc/). Only Illumina or Hybrid (Illumina short and ONT long-reads) de novo genome assemblies were performed using Unicycler v0.4.8[51] (details of the QC stats are provided in Supplementary Data 1c, d)[50]. *Enterobacter* species were identified based on the "overall genome related index" (OGRI)[52] and included 23 *Enterobacter* species valid as of June 2022[53,54]. The OGRI was determined by calculating the average nucleotide identity (ANI) and in silico DNA-DNA hybridization score. The ANI was calculated by BLAST using JSpecies v1.2.1 tool[55]. As the pairwise comparison by JSpecies may differ slightly and is problematic for borderline species[56], we also obtained ANI using the two-way approach from the Enveomics package (http://enve-omics.ce.gatech.edu/enveomics/) (with identical settings of JSpecies v1.2.1 alignment program: blast, window size 1020 bp, step size 150 bp, minimum alignment length 714 bp (70%), minimum alignment identity 306 bp (30%)). The in silico DNA-DNA hybridization score was calculated by genome-to-genome distance calculator (GGDC) using formula-2, respectively[57].

Phylogenomic grouping was carried out using RhierBAPS[58]. The lineages were defined based on the concordance of bootstrap supported phylogenomic grouping and BAPS clusters (assignment probability >0.99). Lineage-specific genes were identified by Scoary v1.6.16[59]. The *hsp60* typing was carried out based on the grouping of partial *hsp60* gene sequence to that of Hoffmann's *hsp60* gene sequence[31]. Phylogenomic analysis for worldwide isolates was performed using MASH v2.1.1[60]. Phylogenomic trees and associated meta-data were visualized in iTOL v4[61]. Genetic structures were visualized in Easyfig v2.1[62]. Plasmid incompatibility groups was performed using PlasmidFinder v2.0.1[63]. Single nucleotide variants (SNVs) were identified by mapping filtered reads against the completely closed ESBL3012 genome (generated in this study) using snippy v4.3.6 (https://github.com/tseemann/snippy). Pairwise genomic comparison and visualization were carried out in Ugene v1.32.0 and the "gene presence/absence" output of Roary, and pan-genome-wide association studies were carried out with Scoary.

Antibiotic resistance gene profiling was carried out using NCBI's AMRfinder (https://www.ncbi.nlm.nih.gov/pathogens/antimicrobial-resistance/AMRFinder/) as well as by "Resistance Gene Identifier" v5.05.5 against CARD database (v3.0.3)[64]. For mobile colistin resistance gene (*mcr*), data on clinical and environmental isolates was compiled from NCBI's Pathogen Detection pipeline (https://www.ncbi.nlm.nih.gov/pathogens/isolates#/search/).

## Comparative analysis of genome sequences of *Enterobacter* spp. obtained worldwide

For comparative genome-based analyses, whole-genome sequences of 3246 non-repetitive isolates (<1000 contigs and >3 Mb assembly size) listed under the genus *Enterobacter* were downloaded from NCBI using e-utilities v2.0 (Supplementary Data 2). These isolates were obtained from clinical, non-clinical, and unknown sources from more than 67 countries. These isolates were selected after reconfirming them as bona fide *Enterobacter* species using OGRI tool (as detailed in Supplementary methods).

## Population analysis profiling (PAP) test

Isolates were grown as described for MIC$_{LB}$. Serial dilutions of culture with O.D.$_{600}$ 0.8–1.0 were prepared and 100 µL culture (from $10^{-4}$, $10^{-6}$, and $10^{-8}$ dilutions) was plated on LB agar plates containing either 0, 8, or 32 mg/L colistin[20]. The plates were incubated at 37 °C for 24 h. CFUs were counted using an automatic colony counter (Scan 500,

Interscience) and heteroresistance frequency was calculated by estimating "CFUs recovered on plate containing colistin" × 100/"CFUs on plate without colistin."

## Mutant generation and complementation

The mutants for *arnBCADTEF*, *pmrAB*, *phoPQ*, and *mgrB* genes were generated by homologous recombination using the λ-Red recombination system[65]. The vector pSIM5-tet carrying the λ-Red genes (γ, β, and exo) was introduced into the strain RBK-17-0394-1[66]. Next, a PCR product for kanamycin resistance gene flanked by FRT-sites (FLP recognition target) was generated from vector pKD4. The transformants carrying the λ-Red recombination system (pSIM5-tet) were grown in LB media containing tetracycline at 30 °C to O.D.$_{600}$ of 0.4. Thermal induction of λ-Red recombination system was carried out by incubating transformant at 42 °C for 15 min. After thermal induction, the transformants were made electrocompetent by washing cells (3 times) with ice-cold 10% glycerol. After electroporation of the linear PCR product, the mutants were selected on agar plates containing 50 mg/L kanamycin at 30 °C. The vector pSIM5-tet was eliminated by growing cells at 42 °C. All the primers (procured from Eurofins genomics) and vectors used are listed in Supplementary Data 3. For complementation, respective genes with their original promoter region (−60 to −100 bp) were PCR-amplified and cloned in a low-copy broad host range vector pBBR1-MCS2. Recombinant plasmids were transformed in the respective mutants. Hybrid *phoP/phoQ* mutants were generated by "Gibson Assembly Cloning Kit" (NEB, #E5510S) following the manufacturer's instructions.

## Lipid A extractions

Lipid A was isolated from freshly harvested bacterial cells using ammonium hydroxide-isobutyric acid extraction[67]. The cells were grown as described for MIC$_{LB}$ testing. At an O.D.$_{600}$ of between 0.8 and 1.0, 5 mL culture were pelleted and resuspended in 400 µL of 70% isobutyric acid and 1 M ammonium hydroxide (5:3, [v/v]) followed by an incubation for 2 h at 100 °C in a screw-cap test tube. The mixture was cooled on ice and centrifuged at 2000 × *g* for 15 min at 4 °C. The resulting supernatant was diluted with water (1:1, [v/v]) and lyophilized (Thermovac TM201; Leybold-Heraeus, Germany). The sample was washed with 1 mL of methanol and centrifuged at 2000 × *g* for 15 min at 4 °C. The insoluble lipid A fraction was solubilized and extracted using 100 µL of a mixture of chloroform:methanol:water (3:1.5:0.25, [v/v/v]) and dried prior to further analysis.

## Mass spectrometry

All mass spectrometric analyses were performed on a Q Exactive Plus (ThermoFisher Scientific, Germany) using a Triversa Nanomate (Advion, USA) as nano-ESI source. Lipid A extracts were initially dissolved in 20 µL chloroform:methanol:water (60:30:5, [v/v/v]) or chloroform:methanol (8:2, [v/v]). 5 µL of such a solution were mixed with 95 µL of water/propan-2-ol/7 M triethylamine/acetic acid (50:50:0.06:0.02, [v/v/v/v]). In those cases where resulting solutions were too viscous for spraying, 1:10 dilutions in water/propan-2-ol/7 M triethylamine/acetic acid (50:50:0.06:0.02, [v/v/v/v]) were used for the measurement. Mass spectra were recorded for 0.50 min in the negative mode in an *m/z*-range of 400-2500 applying a spray voltage set of −1.1 kV. All depicted mass spectra were charge deconvoluted (Xtract module of Xcalibur 3.1 software (ThermoFisher Scientific, Germany)), and given mass values refer to the monoisotopic masses of the neutral molecules, if not indicated otherwise. For the calculation of the ratios of non-modified and L-Ara4N-modified lipid A species mass spectrometric data of 143 acquisitions including 6 blanks were imported into one database using LipidXplorer 1.2.8[68]. Monoisotopic peaks of 57 lipid A species were calculated and assigned. Intensities of all observed tetra- to hepta-acylated lipid A species were considered and utilized to determine substitution ratios. Details on the lipid A molecular species,

detected ions, and calculation of lipid A species relative abundance can be found in Supplementary Data 4.

## Fluorescence microscopy

Bacteria were grown in conditions as used for the $MIC_{LB}$ test. From freshly grown cultures (O.D.$_{600}$ 0.8–1.0), 2 mL culture was pelleted (8000 × $g$ for 10 min) and resuspended in 0.85% NaCl. The O.D.$_{600}$ was adjusted to 0.1 (~$10^8$ CFUs/mL) and two sets (1 mL each) were created as treatment and control. For treatment, colistin sulfate was added to the final concentration of 64 mg/L and incubated at room temperature for 15 min. Cells were pelleted, washed twice, and resuspended in 0.85% NaCl solution. To determine cells with intact or compromised cell membranes, we used LIVE/DEAD™ BacLight™ Bacterial Viability Kit (ThermoFisher, # L7012). Equal volumes of component A (SYTO 9, 3.34 mM) and component B (Propidium iodide, 20 mM) were mixed, and 3 μL of dye mixture was added to 1 mL of cell suspension and incubated for 15 min in dark. Cell membrane integrity was also evaluated using FM4-64 (AAT Bioquest, #21487) (that stain membranes red) combined with DAPI (that following entry stains DNA blue). FM4-64 and DAPI were added to the final concentration of 2 and 1 mg/L, respectively, and incubated for 15 min. Labeled bacterial suspensions (5 μL) were visualized under the fluorescence microscope using Plan Apochromat 60X Oil immersion lens (Keyence Biozero BZ-8000K). Images were merged by BZ analyzer v3.61 with default settings for colors (Keyence). For LIVE/DEAD™ staining, cells with green fluorescence were considered cells with intact membrane, while those with red fluorescence were considered membrane-compromised cells. The live/dead count was carried out using ImageJ v1.53e.

## RNA extraction, cDNA synthesis, and quantitative PCR

Bacteria were grown as described for the $MIC_{LB}$ test. Once the O.D.$_{600}$ of cultures reached to 0.8–1.0, 1 mL of culture was mixed with RNAprotect reagent and stored at −80 °C. For RNA isolation, the mixture was defrosted on ice and pelleted by centrifugation (8000 × $g$, 5 min at 4 °C). The pellet was processed by RNeasy Mini Kit (Qiagen, #74124) following the manufacturer's protocol. The cDNA synthesis was performed using SuperScript II (Invitrogen) using 100 ng of total RNA with random hexamer and nonamer primers. Quantitative PCR amplification was performed by using 1 μL aliquot of a 1st strand cDNA reaction with brilliant SYBR Green qPCR master mix (Qiagen) on the OnePlus real-time cycler (Applied Biosystem) in a final volume of 25 μL. The specificity of all the amplicons was confirmed by gel analysis and melting curves. Experiments were performed in triplicate for three biological replicates. 16S rRNA was used as an endogenous control. Relative quantification was performed using the 2(−ΔΔCt) method[69]. The primers used are listed in Supplementary Data 3.

## Computational modeling of PhoQ and MgrB

To compute interactions between PhoQ sensor protein of *E. xiangfangensis* RBK-17-0394-1 (WT), *E. roggenkampii* RPB-17-0516-2 (516), and *E. bugandensis* RBG-17-0246-1 (246) with MgrB, we used reiterative homology modeling and molecular docking techniques. Homology-based PhoQ models were built using computationally build PhoQ[70], as a template structure (https://salilab.org/phoq) through Swiss-model server[71]. To model the PhoQ periplasmic, trans-membrane and HAMP region 3BQ8.pdb was used for periplasmic domain, 1H2S.pdb was employed for trans-membrane domain, and 2ASW.pdb for HAMP region[70]. The atomic structure of MgrB was modeled using the AlphaFold server[72]. The stereo-chemical qualities of modeled structures of PhoQ and MgrB were evaluated using Ramachandran Plot through the PROCHECK[73]. To explore the interaction of C-terminal periplasmic region of MgrB with periplasmic domain (i.e. acidic region) of PhoQ, molecular docking were performed using HADDOCK[74]. The analysis and visualization of molecular interactions

were performed using the PyMOL Molecular Graphics System, Version 2.0 Schrödinger, LL.

## Reporting summary

Further information on research design is available in the Nature Portfolio Reporting Summary linked to this article.

## Data availability

The genome sequencing data generated in this study have been deposited in the NCBI database under the BioProject accession number PRJNA622426. Experimental data for antibiotic resistance phenotype, minimum inhibitory concentration (MIC) measurements and mass spectrometry are provided in Supplementary Material, Supplementary, or Source data files. Fluorescence microscopy raw images are submitted as a dataset on Zenodo with DOI: 10.5281/zenodo.7405690. All data are available without any restriction. Source data are provided with this paper.

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

## Acknowledgements

We thank Christina Gerstmann, Rachel Javorova (Erasmus visiting student), and Michelle Wröbel for excellent technical assistance. This study was supported by grants from the Bundesministerium für Bildung und Forschung within the framework of the German Center for Infection Research (DZIF) and the Deep iAMR project to T.C. (8032808811, 8032808817, 8032808818, 8032808820, 031L0209B). Research in the D.S. laboratory was supported by the German Center for Infection Research (TTU TB DZIF-709).

## Author contributions

S.P.D. and T.C. conceived research on colistin resistance/hetero-resistance. The clinical study was conducted by the R-Net study group. S.P.D. analyzed genomes and performed the experiments together with R.F. and C.G., and N.G. and D.S. designed and performed the mass spectrometry studies. S.P.D. and R.F. created mutants and performed complementation analysis. B.V.K. carried out protein modeling studies and constructed models. J.F. and C.I. collected isolates, AMR, and other meta-data from partner sites. L.F. carried out the plasmid bioinformatic analysis. Members of consortia, A.M., J.R., M. Behnke, M. Buhl, S.E., P.G., H.G., N.K., W.K., A.K., E.K., S.P., A.R., G.H., H.S., E.T., M.V., S.V.W., and J.Z. provided access to isolates and relevant meta-data for research from respective study sites. C.I., N.G., D.S., and T.C. supervised the work and interpreted the results. T.C. and S.D. wrote the manuscript with inputs from N.G., D.S., and all other authors. T.C. acknowledges assistance from the Shushila and Hoima Podo Chakraborty Foundation during the writing of this manuscript.

## Funding

## Competing interests

The authors declare no competing interests.

## Additional information

[1]German Center for Infection Research (DZIF), Braunschweig, Germany. [2]Institute of Medical Microbiology, Justus Liebig University, Gießen, Germany. [3]Division of Bioanalytical Chemistry, Priority Area Infections, Research Center Borstel, Leibniz Lung Center, Borstel, Germany. [4]Department of Biological Sciences, Sunandan Divatia School of Science, NMIMS (Deemed-to-be) University, Vile Parle, Mumbai, India. [5]Institute of Hygiene and Environmental Medicine, Justus Liebig University, Gießen, Germany. [6]Department of Infectious Diseases and Microbiology, University of Lübeck, Lübeck, Germany. [7]Charité-Universitätsmedizin Berlin, corporate member of Freie Universität Berlin, Humboldt-Universität of Berlin and Berlin Institute of Health, Institute of Hygiene and Environmental Medicine, Berlin, Germany. [8]Institute of Medical Microbiology and Hygiene, Tübingen University, Tübingen, Germany. [9]Division of Infectious Diseases, Department of Internal Medicine I, Tübingen University, Tübingen, Germany. [10]Institute of Clinical Hygiene, Medical Microbiology and Infectiology, Paracelsus Medical University, Klinikum Nürnberg, Nürnberg, Germany. [11]Institute for Medical Microbiology and Hygiene, Albert-Ludwigs-University, Freiburg, Germany. [12]Division of Infectious Diseases, Department of Medicine II, Faculty of Medicine and University Hospital and Medical Center, Albert-Ludwigs-University, Freiburg, Germany. [13]Institute for Medical Microbiology, Immunology, and Hygiene, Faculty of Medicine and University Hospital Cologne, University of Cologne, Cologne, Germany. [14]Department I of Internal Medicine, Faculty of Medicine and University Hospital Cologne, University of Cologne, Cologne, Germany. [15]Department of Internal Medicine, Infectious Diseases, University Hospital Frankfurt, Goethe University Frankfurt, Frankfurt, Germany. [16]Airway Research Center North, Member of the German Center for Lung Research (DZL), Site: Research Center Borstel, Borstel, Germany. [22]These authors contributed equally: Swapnil Prakash Doijad, Nicolas Gisch. ✉e-mail: Trinad.Chakraborty@mikrobio.med.uni-giessen.de

## DZIF R-Net Study Group

L.A. Peña Diaz[17], G. Pilarski[17], N. Thoma[17], A. Weber[17], M. Vavra[18], S. Schuster[18], G. Peyerl-Hoffmann[18], A. Hamprecht[18], S. Proske[19], Y. Stelzer[19], J. Wille[19], D. Lenke[20], B. Bader[21], A. Dinkelacker[21], F. Hölzl[21] & L. Kunstle[21]

[17]Berlin, Germany. [18]Freiburg, Germany. [19]Köln, Germany. [20]Lübeck, Germany. [21]Tübingen, Germany.

