## [Peer review file · Nature Communications]

REVIEWER COMMENTS

Reviewer #1 (Remarks to the Author):

The authors present a comprehensive study of colistin resistance and heteroresistance, with covering many different aspects of development of resistance, making use of genomic information as well as laboratory techniques to uncover the basic mechanism.

The study is very relevant, and the methods sound and appropriate.

A couple of further questions and comments.

Line 142-148: Please make the ONT data available as well. As far as I can see in the ENA, only ILM data has been submitted under study PRJNA622426.

Lines 150-151: I cannot find any mention of quality control or which assembler was used in the supplementary data.

Lines 161-179: You mention several ways of phylogenomic grouping, however, you only present the phylogeny based on core genes. Perhaps in the interest in keeping the materials and methods section neat and tidy, it would be best to move the other methods to the supplementary data discussion, and then do mention that you tried several methods that all agree. This seems to be missing in the main as well as the supplementary text. The supplementary data mentions various ANI approaches, but not SNP-based phylogeny nor the core genome by Harvest suite approach.

Lines 180-184: Similarly to the phylogenomic grouping, some of these methods seem to be redundant, as there is no mention of phages, or virulence factors in the manuscript, and identification of plasmids only occurs with respect to the position of the *mcr-9* gene.

Lines 557-565: I find it slightly strange to introduce new results in the discussion section. Perhaps this should be in the results section, with a mention in the discussion?

Figure 1b: please check your colour scheme for the genes displayed. I can only see the *arn* gene cassette colour in *xiangfangensis*, in the other species, this colour is different, but does not correspond to the “*iro* gene cassette” which is also not mentioned anywhere in the main text.

Figure 1c: This figure does not add much in my opinion. Also, what is your current cutoff for displaying countries? This is not apparent, and Turkey with 6 isolates, 100% *xiangfangensis* is displayed, but Morocco with 4 isolates (100%) is not. In both cases, I would argue that the number of isolates is too small to draw a valid conclusion. It is also skewed very much by which countries submit sequenced isolates to NCBI. I would suggest removing the figure and refer to supplementary table 2 to show the worldwide distribution of *xiangfangensis*, which holds true.

Figure 2c: legend: “*Enteorbcter*”. Further to Figure 2c and d: How representative are the levels of modified L-Ara4N if only a single representative of each species was measured? And is this then truly “species-dependent levels of lipid A modification” as stated in the text (Lines 382-383). I guess that

some might hold true as shown in the PhoQ complementation studies for xiangfangensis, but this could be related back to the variation in phoQ, and should be put in context.

Supplementary Table 6: You might want to change green to blue, to make it colour-blind friendly.

Reviewer #2 (Remarks to the Author):

Resolving colistin resistance and heteroresistance in Enterobacter species

Authors: Swapnil Prakash Doijad¹, Nicolas Gisch, Renate Frantz, Bajarang Vasant Kumbhar, Jane Falgenhauer, Can Imirzalioglu, Linda Falgenhauer, Alexander Mischnik, Jan Rupp, Michael Behnke, Michael Buhl, Simone Eisenbeis, Petra Gastmeier, Hanna Gölz, Georg Alexander Häcker, Nadja Käding, Winfried V. Kern, Axel Kola, Evelyn Kramme, Silke Peter, Anna M. Rohde, Harald Seifert, Evelina Tacconelli, Maria J.G.T. Vehreschild, Sarah V. Walker, Janine Zweigner, Dominik Schwudke and Trinad Chakraborty on behalf of the DZIF R-Net Study Group

In this study, authors have performed a genome-based taxonomic study on clinical isolates of Enterobacter obtained over a three-year period from six university hospitals at different locations in Germany.

General comments:

It would be advisable to review the introduction as a whole. Some important bibliographic data (doi: 10.1093/jac/dkz236.) are missing to understand the mechanisms of colistin resistance or heteroresistance. The determination of colistin MICs does not comply with EUCAST recommendations (no reference strain, too high inoculum...) On the other hand, many data have already been described. On the other hand, the genomic approach is better exploited and well described. It would be desirable to revise the whole article in this sense. The quality of the figures is remarkable

Major comments:

- Introduction:

o Lines 88-94: colistin resistance in enterobacter is cluster dependent? this has already been described? this notion does not appear in the introduction?

o Introduction: why not also introduce the importance of ecr in colistin resistance? doi: 10.1093/jac/dkz236.

- Methods:

o Lines 125-134 : why did you use LB broth instead of cation-adjusted MH for colistin MIC determination ? this is not the method recommended by EUCAST?

o As a reminder, the recommendations state:

a. Cation-adjusted Mueller-Hinton Broth is used

b. No additives may be included in any part of the testing process (in particular, no polysorbate-80 or other surfactants)

c. Trays must be made of plain polystyrene and not treated in any way before use

d. Sulphate salts of polymyxins must be used (the methanesulfonate derivative of colistin must not be used - it is an inactive pro-drug that breaks down slowly in solution)

o What reference strains did you use to ensure MIC results?

o the inoculum used to perform MICs is at least 100 times higher than a recommended inoculum? how can you justify the MIC results obtained, especially for all the strains belonging to the *E. xiangfangensis*

o Lines 166-168 and 501-512: why not use the *dnal* gene for clustering (DOI : 10.1128/Spectrum.01242-21)?

-

Minor comments :

- Lines 410-426: this part is not interesting, already described in many articles: ex doi: 10.1093/jac/dkw260, doi: 10.1128/AAC.00237-16.

- Line 895: please replace “*Enteorbcter species*” by “*Enterobacter species*”

☒ What are the noteworthy results? The results of the genomic analysis are very interesting and well presented. But these results are difficult to use with the phenotypic approach (MIC determination)

☒ Will the work be of significance to the field and related fields? How does it compare to the established literature? If the work is not original, please provide relevant references : This work is original on the genomic level but not at all on the mechanistic level of colistin resistance. Some major references are missing.

☒ Does the work support the conclusions and claims, or is additional evidence needed? The determination of MICs for colistin raises many questions as to whether the recommendations are being followed correctly.

☒ Are there any flaws in the data analysis, interpretation and conclusions? Do these prohibit publication or require revision? This article requires clarification on the determination of MICs for colistin. The introduction is incomplete. A major revision is needed to confirm the conclusions obtained and the very interesting genomic approach

☒ Is the methodology sound? Does the work meet the expected standards in your field? no

☒ Is there enough detail provided in the methods for the work to be reproduced? yes

Reviewer #1 (Remarks to the Author):

We thank the reviewer for his overall generous assessment of the manuscript.

Questions and comments:

Line 142-148: Please make the ONT data available as well. As far as I can see in the ENA, only ILM data has been submitted under study PRJNA622426.

- The ONT data is now publicly available.

Lines 150-151: I cannot find any mention of quality control or which assembler was used in the supplementary data.

- The quality control and assembly data are now added in the method section and reads as follows:
“For bioinformatic analysis, default parameters were used for all software unless otherwise specified. Illumina raw reads were filtered using Trimmomatic v0.36. ONT long reads demultiplexing and adapter trimming were performed using Porechop v0.2.4 (<https://github.com/rrwick/Porechop>). The reads were examined by FastQC (<https://www.bioinformatics.babraham.ac.uk/projects/fastqc/>). Only Illumina or Hybrid *de novo* genome assemblies were performed using Unicycler v0.4.8 (details of the QC statistics are provided in Supplementary data 1).”

Lines 161-179: You mention several ways of phylogenomic grouping, however, you only present the phylogeny based on core genes. Perhaps in the interest in keeping the materials and methods section neat and tidy, it would be best to move the other methods to the supplementary data discussion, and then do mention that you tried several methods that all agree. This seems to be missing in the main as well as the supplementary text. The supplementary data mentions various ANI approaches, but not SNP-based phylogeny nor the core genome by Harvest suite approach.

- As a primary approach, we used clustering based on concatenated core genes combined with Bayesian Analysis of Population Structure (BAPS). This information is now available in the main text. A description of other phylogenomic approaches with comparative phylogenies is now presented in the supplementary material as new Supplementary Figure 1. The information deriving from other phylogenomic approaches has been inserted into the paragraph providing description of the lineages.

Lines 180-184: Similarly, to the phylogenomic grouping, some of these methods seem to be

redundant, as there is no mention of phages, or virulence factors in the manuscript, and identification of plasmids only occurs with respect to the position of the *mcr-9* gene.

- These redundant and irrelevant information has now been removed.

Lines 557-565: I find it slightly strange to introduce new results in the discussion section. Perhaps this should be in the results section, with a mention in the discussion?

- This data is now presented in the results section of the amended manuscript.
- Data based on MLST typing and clones are now also included in the results section of the amended manuscript.
- Backward compatibility data relating to *hsp60*-based typing is reflected in Figure 1b, the discussion section, and supplementary table 5. This redundant description is removed.
- Further information regarding the taxonomic classification is now presented in the legend to Supplementary Table 1.
- This section has been re-titled as “High occurrence of the *mcr-9* gene in *E. xiangfangensis*” and now references both the antibiotic resistance gene profiles and a description of the overall genetic environment of *mcr-9*.

Figure 1b: please check your colour scheme for the genes displayed. I can only see the *arn* gene cassette colour in *xiangfangensis*, in the other species, this colour is different, but does not correspond to the “*iro* gene cassette” which is also not mentioned anywhere in the main text.

- The color scheme has been updated. The description on the *iro* gene cassette has been removed.

Figure 1c: This figure does not add much in my opinion. Also, what is your current cutoff for displaying countries? This is not apparent, and Turkey with 6 isolates, 100% *xiangfangensis* is displayed, but Morocco with 4 isolates (100%) is not. In both cases, I would argue that the number of isolates is too small to draw a valid conclusion. It is also skewed very much by which countries submit sequenced isolates to NCBI. I would suggest removing the figure and refer to supplementary table 2 to show the worldwide distribution of *xiangfangensis*, which holds true.

- A cutoff of minimum 5 isolates per country was used. We agree that the public data is subjective with respect to the isolates sequenced and deposited. As suggested, Figure 1c is now removed to avoid any bias, and an overall description of the worldwide data is presented the Supplementary table 2.

Figure 2c: legend: “Enteorbcter”. Further to Figure 2c and d: How representative are the levels of modified L-Ara4N if only a single representative of each species was measured? And is this then truly “species-dependent levels of lipid A modification” as stated in the text (Lines 382-383). I guess that some might hold true as shown in the PhoQ complementation studies for

xiangfangensis, but this could be related back to the variation in phoQ, and should be put in context.

- Spelling of *Enterobacter* has been corrected.
- By way of explanation, we first determined heteroresistance frequencies (HRFs) towards colistin using the population analysis profiling (PAP) assay (at two different concentrations i.e at 8- and 32 mg/L) for all 165 isolates of *Enterobacter* used in this study (Figure 2D). Isolates were clustered to the level of species using genome-based taxonomy as described above. We selected isolates whose heteroresistance to colistin (at both concentrations) were representative of individual species. The quantitative lipidomics data of extracted lipid A from these isolates presented this study is the result of statistical analysis deriving from at least three (and up to six) independent experiments performed for every isolate tested. We found a very strong positive correlation between the proportions of L-Ara4N-modified lipid A and their HRF (Pearson correlation coefficient: 0.9 and 0.72 for 8 and 32 Mg/L respectively)
We appreciate the caveat raised by the reviewer and have qualified our findings to read: “We detected varying levels of L-Ara4N modification of lipid A in individual isolates representing different species (Figure 2c).”

Supplementary Table 6: You might want to change green to blue, to make it colour-blind friendly.

- Changed from green to blue, as suggested.

Reviewer #2 (Remarks to the Author):

General comments:

It would be advisable to review the introduction as a whole. Some important bibliographic data (doi: 10.1093/jac/dkz236.) are missing to understand the mechanisms of colistin resistance or heteroresistance. The determination of colistin MICs does not comply with EUCAST recommendations (no reference strain, too high inoculum...) On the other hand, many data have already been described. On the other hand, the genomic approach is better exploited and well described. It would be desirable to revise the whole article in this sense. The quality of the figures is remarkable

- Following the overall comments of the reviewer we have now added a paragraph in the introduction to include suggestions made.

“*Enterobacter* isolates frequently exhibit heteroresistance towards colistin making accurate resistance testing difficult^{19,20} leading to a high risk of treatment failures particularly with those isolates that have previously been classified as susceptible^{19,21}. Previous studies have implicated the two component systems (TCSs) PhoPQ/PmrAB that regulate expression of the *arnBCADTEF* gene cassette^{22,23}, the PhoPQ inhibitor- MgrB and enhancer- Ecr peptides²⁴, the inner membrane protein DedA²⁴, and AcrAB-TolC efflux pump²⁵ with colistin heteroresistance in different species of *Enterobacter*. Recent studies indicate that the TCS PmrAB is not involved in heteroresistance towards colistin in *E. cloacae*^{20,22}. In addition, taxonomic conflicts in ECC have further complicated efforts to consistently identify genetic features underlying heteroresistance at species level. Membership based on a *hsp60* gene classification scheme first provided clues for distinct cluster-dependent heteroresistance levels and sensitive populations within ECC^{20,26}. However, heteroresistance frequencies varied greatly within specific clusters suggesting mutations in additional genes or allelic differences underly these findings.”

Major comments:

- Introduction:

o Lines 88-94: colistin resistance in enterobacter is cluster dependent? this has already been described? this notion does not appear in the introduction?

- The *hsp60*-based cluster-dependent colistin heteroresistance was first described by Guérin et al. (2016) and used more recently by Pantel et al. (2022). We have acknowledged these important findings in the introduction (at the end of the third paragraph):
“ Membership based on a *hsp60* gene classification scheme first provided clues for distinct cluster-dependent heteroresistance levels and sensitive populations within ECC^{20,26}”

o Introduction: why not also introduce the importance of *ecr* in colistin resistance? doi: 10.1093/jac/dkz236.

- We thank the reviewer for the suggestion that we have now incorporated information regarding *Ecr* into the introduction section of the revised manuscript (see above). We also performed an analysis for the presence/absence of the *ecr* gene in publicly available *Enterobacter* genomes (n= 3246). The *ecr* gene is absent from the genome of the most commonly occurring species (>66%) detected in this study viz., *E. xiangfangensis* (all lineages). Other species lacking this gene include *E. hormachaei*, *E. oligotrophicus*, and *E. wuhensis*. The data derived from this analysis is provided in the Table below for the reviewer. Data regarding the presence/absence of *ecr* in the individual isolates is now provided for perusal and listed in Supplementary data files 1 and 2).

OGRI_based_species_(ANI>95%)	total isolates	ecr+ve (%)
E. hormaechei	31	0 (0.0)
E. oligotrophicus	1	0 (0.0)
E. timonensis	1	0 (0.0)
E. wuhouensis	22	0 (0.0)
E. xiangfangensis L-4	152	0 (0.0)
E. xiangfangensis L-3	740	0 (0.0)
E. xiangfangensis L-2	464	0 (0.0)
E. xiangfangensis L-1	974	0 (0.0)
Unidentified Enterobacter species-1	52	3 (5.8)
E. dykesii	5	4 (80.0)
E. vonholyi	14	13 (92.9)
E. asburiae	205	203 (99.0)
E. roggenkampii	152	151 (99.3)
E. bugandensis	72	72 (100.0)
E. cancerogenus	15	15 (100.0)
E. chengduensis	23	23 (100.0)
E. chuandaensis	4	4 (100.0)
E. cloacae	162	162 (100.0)
E. huaxiensis	2	2 (100.0)
E. kobei	144	144 (100.0)
E. ludwigii	83	83 (100.0)
E. mori	10	10 (100.0)
E. quasihormaechei	1	1 (100.0)
E. quasimori	1	1 (100.0)
E. quasiroggenkampii	16	16 (100.0)
E. sichuanensis	19	19 (100.0)
E. soli	4	4 (100.0)

E. tabaci	1	1 (100.0)
Unidentified Enterobacter species -10	2	2 (100.0)
Unidentified Enterobacter species -11	1	1 (100.0)
Unidentified Enterobacter species -2	17	17 (100.0)
Unidentified Enterobacter species -3	5	5 (100.0)
Unidentified Enterobacter species -4	8	8 (100.0)
Unidentified Enterobacter species -5	8	8 (100.0)
Unidentified Enterobacter species -6	2	2 (100.0)
Unidentified Enterobacter species -7	1	1 (100.0)
Unidentified Enterobacter species -8	1	1 (100.0)
Unidentified Enterobacter species -9	1	1 (100.0)

- Methods:

o Lines 125-134 : why did you use LB broth instead of cation-adjusted MH for colistin MIC determination ? this is not the method recommended by EUCAST?

o As a reminder, the recommendations state:

- a. Cation-adjusted Mueller-Hinton Broth is used
- b. No additives may be included in any part of the testing process (in particular, no polysorbate-80 or other surfactants)
- c. Trays must be made of plain polystyrene and not treated in any way before use
- d. Sulphate salts of polymyxins must be used (the methanesulfonate derivative of colistin must not be used - it is an inactive pro-drug that breaks down slowly in solution)

- In our initial studies we determined MICs using Cation Adjusted Mueller-Hinton Broth following recommendations as laid down by EUCAST (a-d). We found that several *E. xiangfangensis* Lineage 1 isolates classified as colistin-susceptible (see Table below) despite their genomic similarity to the other colistin-resistant isolates from this specific lineage. MIC determinations using LB as a growth medium removed this uncertainty and provided excellent genotype-phenotype correlations. Now, we have provided data for the MIC determined following EUCAST approach for all the 165 isolates in the Supplementary data 1.
- Following on from the suggestion by the reviewer, we performed comparative MIC determinations, to contrast and highlight differences in colistin-susceptibility using either growth media. This data is presented below and is provided for scrutiny by the reviewer.

Spp	Isolate	Colistin MIC range				
		LB media			MHB2	
		inoculum O.D.600 0.8-1.0	inoculum O.D.600 0.08-0.1	pH5, inoculum O.D.600 0.8-1.0	inoculum O.D.600 0.08-0.1	pH5, inoculum O.D.600 0.08-0.1
E_xiangfangensis L-1	BK4767	32	4-8	256	1	256-512
E_xiangfangensis L-1	BK6928	16	4-8	512	4	64-256
E_xiangfangensis L-1	RBK-17-0230-1	4-16	32	256	8	256
E_xiangfangensis L-1	RBK-18-0386-2	4	16	128	0.5	128
E_xiangfangensis L-1	RPB-18-0140-1	64	32	256	16	256
E_mori	RBL-17-0354-2	32-64	32	256-512	8	256
E_roggenkampii	RBK-17-0344-2	>512	256-512	512	64-256	128-512
E_roggenkampii	RPB-17-0516-2	64->128	256-512	>512	512	256-512
E_roggenkampii	RPG-17-0579-1	>512	256	>512	32-64	>512
E_ludwigii	And4961	128	64	512	16	>512
E_ludwigii	BK6227	64	32-64	256-512	128	256
E_ludwigii	F-8789	8-64	16-32	512	16	>512
E_kobei	BK6664	32	256-512	128	32	128
E_kobei	RFB-17-0338-1	>512	>512	256	256	256-512
E_kobei	RBG-16-0046-1	>512	>512	>512	>512	>512
E_kobei	RBL-16-0092-1	8	64-256	16	64-256	16-64
E_cloacae	ESBL2036	>512	>512	>512	>512	>512
E_cloacae	RFB-17-0514-1	512	64	>512	16	512
E_cloacae	RBL-17-0228-1	>512	>512	512	>512	512
E_chengduensis	RBK-18-0141-1	64-128	64-128	512->512	32	256-512
E_cancerogenus	RPK-18-0479-1	64-128	32-64	512->512	32	512
E_bugandensis	BK7261	>512	>512	>512	>512	256-512
E_bugandensis	F-1367	>512	>512	>512	>512	512
E_asburiae	BK5433	256	128	>512	128	>512
E_asburiae	RBG-18-0415-1	128-256	128-256	512	128	256-512

o What reference strains did you use to ensure MIC results?

- *E. coli* ATCC 25922 as a reference strain for MIC determination. The strain exhibited 1-2 mg/L MIC towards colistin. This information is now provided.

o the inoculum used to perform MICs is at least 100 times higher than a recommended inoculum? how can you justify the MIC results obtained, especially for all the strains belonging to the *E. xiangfangensis*

Following the reviewer's suggestion, we have performed MIC determinations for all isolates in this study following standard EUCAST protocols. This information is now presented in supplementary data 1 and mentioned in the Discussion:

"To correlate lipid A profiles, heteroresistance frequency and data from microscopic studies, we determined MICs using identical growth conditions i.e., using LB broth. When compared to MIC levels obtained with standard EUCAST protocols, the MIC_{LB} values were either similar or two-fold higher (supplementary data 1b). As with the EUCAST approach the "skipped well phenomenon" was also observed during MIC_{LB} determinations."

o Lines 166-168 and 501-512: why not use the *dnaJ* gene for clustering (DOI : 10.1128/Spectrum.01242-21)?

- We performed *dnaJ* gene-based clustering for all 165 isolates presented in this study following the suggestion of the reviewer. We found, with a single exception, an excellent correlation to the genome-based taxonomy provided here, corroborating the validity of the findings presented here. This data is now provided below as an additional figure for the reviewer.

We note that our backward compatibility studies are based on comparative analysis of our genome-based taxonomy to a large set of data obtained with *hsp60* gene-based clustering ((Hoffmann et al., 2003; Paauw et al., 2008, Morand et al., 2009; Guérin et al., 2016; Chavda et al., 2016, Moradigaravand et al., 2016; Garinet 2018 and Peirano et al., 2018, Pantel et al., 2022). As the *dnaJ*-based scheme developed is relatively new, backward compatibility studies are more limited, but warrant its use in further studies.

We envisage that the the generation of species-specific profiles – using MALDI-TOF and based on the taxonomic scheme described here – would provide for rapid and robust species designations in clinical microbiology laboratories in the near future.

Minor comments:

- Lines 410-426: this part is not interesting, already described in many articles: ex doi: 10.1093/jac/dkw260, doi: 10.1128/AAC.00237-16.

- We have now removed the title of the section and integrated this data into a single section titled:

“Allelic differences in *phoQ* and *mgrB* determines colistin resistance and heteroresistance”

- Line 895: please replace “Enteorbcter species” by “Enterobacter species”

- Replaced as suggested.

REVIEWERS' COMMENTS

Reviewer #2 (Remarks to the Author):

Resolving colistin resistance and heteroresistance in *Enterobacter* species

Authors: Swapnil Prakash Doijad¹, Nicolas Gisch, Renate Frantz, Bajarang Vasant Kumbhar, Jane Falgenhauer, Can Imirzalioglu, Linda Falgenhauer, Alexander Mischnik, Jan Rupp, Michael Behnke, Michael Buhl, Simone Eisenbeis, Petra Gastmeier, Hanna Gölz, Georg Alexander Häcker, Nadja Käding, Winfried V. Kern, Axel Kola, Evelyn Kramme, Silke Peter, Anna M. Rohde, Harald Seifert, Evelina Tacconelli, Maria J.G.T. Vehreschild, Sarah V. Walker, Janine Zweigner, Dominik Schwudke and Trinad Chakraborty on behalf of the DZIF R-Net Study Group

General comments:

The corrections of this article were perfectly conducted by the authors. The answers are satisfactory and allow for the completion of the data.

The article is well constructed with an excellent scientific argument. The bibliographical references are in accordance with the subject and well updated

The quality of the figures is remarkable